

# Conservation of heat and mass in P-SKRIPS version 1: the coupled atmosphere-ice-ocean model of The Ross Sea

Alena Malyarenko[1,2,*], Alexandra Gossart[2,*], Rui Sun[3], and Mario Krapp[4,2]

[1]National Institute of Water and Atmospheric Research | Taihoro Nukurangi , 301 Evans Bay Parade, Hataitai, Wellington, New Zealand
[2]Antarctic Research Centre | Te Puna Pātiotio, Victoria University of Wellington, CO 549, Cotton Building, Gate 7, Kelburn Parade, Wellington, 6012, New Zealand
[3]Scripps Institution of Oceanography, La Jolla, California, USA
[4]GNS Science | Te Pū Ao, 30 Gracefield Rd, Gracefield, Lower Hutt 5010, New Zealand
[*]These authors contributed equally to this work.

**Correspondence:** Alena Malyarenko (Alena.Malyarenko@niwa.co.nz) or Alexandra Gossart (alexandra.gossart@vuw.ac.nz)

**Abstract.**

Ocean-atmosphere-sea ice interactions are key to understanding the future of the Southern Ocean and the Antarctic continent. Regional coupled climate-sea ice-ocean models have been developed for several polar regions, however the conservation of heat and mass fluxes between coupled models is often overlooked due to computational difficulties. At regional scale, the non-
conservation of water and energy can lead to model drift over multi-year model simulations. Here we present P-SKRIPS version 1, a new version of the SKRIPS coupled model set up for the Ross Sea region. Our development includes a full conservation of heat and mass fluxes transferred between the climate (PWRF) and sea ice-ocean (MITgcm) models. We examine open water, sea ice cover, and ice sheet interfaces. We show the evidence of the flux conservation in the results of a one month-long summer and one month-long winter test experiments. P-SKRIPS v.1 shows the advantages of conserving heat flux over the Terra Nova
Bay and Ross Sea polynyas in August 2016, eliminating the mismatch between total flux calculation in PWRF and MITgcm up to 922 W m$^{-2}$.

## 1   Introduction

The Southern Ocean, the sea ice it produces, and the Antarctic continent play a central role in the global climate system. Antarctica's coastal margins are sensitive to changes in any of the components of the system; there the glacier ice, the ocean,
the atmosphere, and the sea ice come together. Ice-ocean-atmosphere interactions and feedbacks are crucial to our understanding of the present state of the Southern Ocean and Antarctica, if we want to be able to predict how the system changes under different conditions. For example, the Southern Ocean is the main source of moisture for the Antarctic continent (van Wessem et al., 2018; Agosta et al., 2019) and it controls the mass gain of land-locked ice and thus global sea level (e.g. Holland et al., 2020; Golledge et al., 2015; Krinner et al., 2007). How much and how far the moisture reaches into the continent is controlled
by sea surface conditions (Kittel et al., 2018), such as sea ice extent (e.g. Bromwich, 1988; Wu et al., 1996; Wang et al., 2020). Antarctic mass loss is controlled by the atmosphere and the ocean. Ice shelves, the gatekeepers of glacial outflow from the con-



tinent (Hubbard et al., 2016; Kingslake et al., 2017; Bell et al., 2018; Scambos et al., 2000; Banwell et al., 2013; Rignot, 2004; Leeson et al., 2020) are affected by the atmospheric and the oceanic forcings. The ponding of surface meltwater (attributed to atmospheric warming) can cause hydrofracturing of ice shelves (e.g. Rott et al., 1996; Scambos et al., 2000; van den Broeke,

2005). Basal melting of ice shelves occurs due to contact with warmer oceanic masses (Etourneau et al., 2019; Shepherd et al., 2004) and can be impacted by the presence of sea ice. In turn, the extent of the sea ice cover is linked to ocean and atmosphere changes (Christie et al., 2022). Yet, despite the relevance of these ice-ocean-atmosphere interactions for the global climate system, Antarctica's coastal margin and the Southern Ocean are still largely unexplored. Observational data is sparse both spatially and temporally, due to the remoteness and harsh conditions prevailing in the Antarctic.

Earth System Models (ESMs) present an alternative to explore the past, present and future state of Antarctica and the Southern Ocean. Studies using Global Circulation Models or Earth System Models have highlighted the importance of atmosphere-ocean-sea ice interactions in the representation of polar systems, especially for estimating the evolution of Antarctica in a future, warmer setting (e.g. Goosse et al., 2018). Warmer mean surface temperatures lead to increased basal melting of ice shelves (Naughten et al., 2021). In addition, warmer ocean water masses in cavities can be induced by increased surface stress

due to thinning of the sea ice (Hellmer et al., 2012) and increased incoming radiation causing surface melting. The latter can lead to ice shelf fragilisation and potential collapse (DeConto and Pollard, 2016).

ESMs that are part of the Coupled Model Intercomparison Project (CMIP) experiments, have coupled global ocean, atmosphere, land and sea ice models (Meehl et al., 1997), but their spatial resolution is rather coarse. In addition, their global physics are not optimised for polar areas (Agosta et al., 2015), leading to various performances in the Arctic and Antarctic.

Although there is a general improvement since previous experiments, the CMIP6 (Eyring et al., 2016) models still represent an ocean surface that is too warm and too fresh for the Southern Ocean, with a too small annual sea ice extent (Beadling et al., 2020). The models struggle to represent realistic sea ice cover (Mohrmann et al., 2021) and show a wide spread in the different terms of sea ice formation/dissipation, leading to large uncertainties in the sea ice budget across the different models (Li et al., 2021; Roach et al., 2020).

In contrast, Regional Climate Models are applied at the regional scale, at higher resolution, and can be adapted to represent the local atmospheric conditions over the ice sheets in a more accurate way, by physically and dynamically downscaling the global boundary conditions. Several regional models have been optimised to specifically represent the polar climate (RACMO (van Wessem et al., 2018), MAR (e.g. Fettweis et al., 2017; Kittel et al., 2018; Agosta et al., 2019), HIRHAM ((Lucas-Picher et al., 2012), COSMO-CLM (Souverijns et al., 2019), PWRF (Bromwich et al., 2013; Hines and Bromwich, 2008)) or the Southern

Ocean (MITgcm (Nakayama et al., 2017; Naughten et al., 2022), ROMS (Richter et al., 2021), ACCESS-OM (Morrison et al., 2020; Moorman et al., 2020), NEMO (Madec et al., 2017)). However, fully coupled ocean-atmosphere-sea ice (and ice sheets) are rare for any of the regional models that have been set up and parameterized for polar regions. As a result, climate and/or ocean models are usually used in standalone mode (forced by external boundary conditions such as reanalyses or future climate projections), and are therefore unable to reflect the potential feedback between the different components. This is problematic

for Antarctica and the Southern Ocean, where the atmosphere, the ocean and the sea ice interact and impact each other in complex ways. A few groups have undertaken the task of coupling some - or all - of the components over the Antarctic ice sheet




or smaller domains (see Table **??**). The TANGO model coupled the atmosphere MAR model to the NEMO-LIM ocean-sea ice models over the Ross Sea sector (Jourdain et al., 2011), and was used to investigate the feedbacks due to heat and moisture fluxes in polynyas. The HIRHAM-NAOSIM 2.0 (Dorn et al., 2018, 2019) evaluates the HIRHAM climate model coupled to

the NAOSIM ocean-sea ice model over the Arctic. Within the PARASO framework, Pelletier et al. (2022) aims at building a fully-coupled model using the suite of regional models: f.ETISh1.7 (ice sheet), NEMO3.6 (ocean), LIM3.6 (sea ice) and COSMO5.0 coupled to CLM4.5 (atmosphere-land).

Two frameworks have recently been developed to couple the atmosphere Polar Weather and Research Forecasting Model

(PWRF) and the ocean Massachusetts Institute of Technology general circulation model (MITgcm) (which includes a sea ice component). The Skripps-KAUST Regional model was applied to the Red Sea (Sun et al., 2019) and the Southern Ocean and Antarctica (Cerovečki et al., 2022). The Arctic regional coupled sea-ice–ocean–atmosphere model (ArcIOAM) was applied to the Arctic sea ice (Ren et al., 2021). Despite their rather good performance, these models nevertheless do not fully balance the heat and freshwater fluxes between the ocean and atmosphere components. This is critical to avoid model drift and inconsis-

tencies between the two model components over long term simulations.

In this paper, we present P-SKRIPS version 1 (P-SKRIPS hereafter), an improved version of the Scripps-KAUST Regional Integrated Prediction System (SKRIPS, (Sun et al., 2019) model, in a set up for the Ross Sea domain. We aim to improve the framework described in Cerovečki et al. (2022) by conserving the heat, freshwater and mass fluxes between PWRF and MITgcm, and compare the simulation results with those obtained using the set up of the previous model. The Ross Sea is

an important location of ice-atmosphere-ocean interactions where relatively small-scale features have an over-proportionate effect on the global ocean circulation: up to $40\%$ of Antarctic Bottom Water is produced there (Orsi et al., 2002). Therefore, conservation of heat and mass fluxes in the Ross sea is crucial for quantifying air-ice-ocean interactions, now and in the future. The paper is organised as follows: In the next section, we present the set up of these regional models and describe the different simulations carried out in this study. The results and main findings will be highlighted in the third section. A discussion and

the final conclusions will be given in section 4.

## 2    Methods

### 2.1    The ocean and sea ice model: MITgcm

We use a regional Ross Sea configuration of the Massachusetts Institute of Technology general circulation model (MITgcm checkpoint 67m) with dynamic/thermodynamic sea ice (Losch et al., 2010) and thermodynamic ice shelf (Losch, 2008) devel-

opments (https://github.com/MITgcm/MITgcm/tree/checkpoint67m). The domain is set up using a polar stereographic projection, and the horizontal domain has 210 x 240 grid cells with approximately 10 km horizontal resolution, generated by PWRF (Figure 1). It has 70 vertical levels focused on the continental shelf on the Ross Sea and ranges from 10m near the surface, 25-50m mid-depth to 250m at the deepest part of the domain. The model bathymetry, ice shelf draft, and grounding line are extracted from BEDMAP-2 product (Fretwell et al., 2013). The coastline has been smoothed to avoid sea ice piling up and has



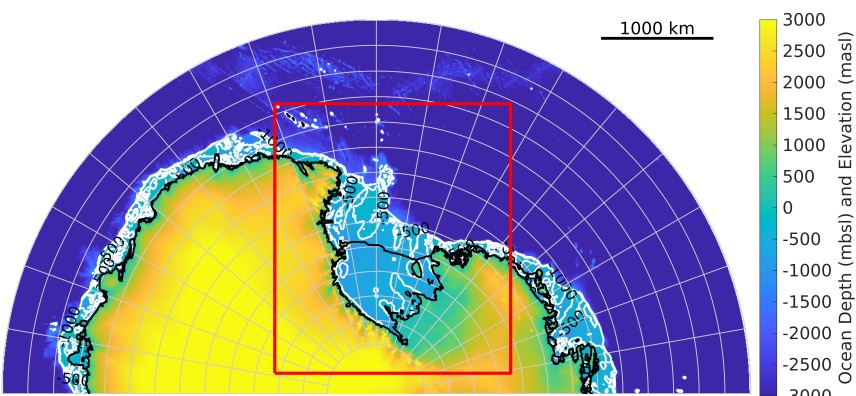

**Figure 1.** Antarctic map with the domain of this study delimited in red. Over open ocean and within ice shelf cavities, colours represent the ocean bathymetry (IBCSO (Arndt et al., 2013)). Over grounded ice sheets, colours represent the surface elevation (topography, Bedmap2). Figure generated with the Antarctic Mapping Toolbox (Greene et al., 2017).

been passed on to PWRF to coordinate the land masks in both models (see Figure 2).

The model initial and boundary conditions have been obtained from biogeochemical-sea ice-ocean state estimate B-SOSE, iteration 135 (Mazloff et al., 2010). The lateral boundary condition of the model is forced with 5-day mean fields: temperature, salinity, velocities, sea ice thickness, sea ice area, sea ice velocities, and snow thickness over sea ice. The ice shelf cavity is filled in by bringing the last available coastal profile southward. No spin up was conducted for this paper as we focus on the

technical aspects of the coupling and the aim is not to match the observed state of the Ross Sea.

## 2.2 The atmosphere model: PWRF

The Weather Research and Forecasting Model (WRF, (Skamarok et al., 2019)) is the atmospheric component of the coupled model. It is a non-hydrostatic mesoscale atmospheric model developed by the NCAR (National Centres for Atmospheric Research) for both atmospheric research and operational forecasting applications. In this study, we use the Polar-optimised version

(PWRF) of the Advanced Research WRF dynamic version 4.1.3 (https://github.com/wrf-model/WRF/releases/tag/v4.1.3). This optimisation has been used to specifically represent the polar climate of the Arctic and Antarctic (Bromwich et al., 2013; Hines and Bromwich, 2008). It enables the sea ice mask to be updated during the simulation, and the associated sea ice thickness and snow on sea ice values to be specified using an external field. The land component of PWRF is the community Noah land surface model with multi-parameterization options (Noah-MP, (Niu et al., 2011)) and dissociates between bare ground, land

ice (glaciers) and sea ice. The latest version includes a three-layer snow model ((Yang and Niu, 2003)). It improves the representation of surface fluxes, surface meltwater production, percolation and retention/refreezing in the snow layers and surface runoff, and has a closer snow water equivalent reproduction with an improved diurnal cycle of the snow skin temperature.

In this study, PWRF is used over the model domain at approximately 10km horizontal resolution on a polar stereographic grid. It has 210 x 240 grid cells and 61 model levels. The static input data come from the Reference Elevation Model of Antarctica



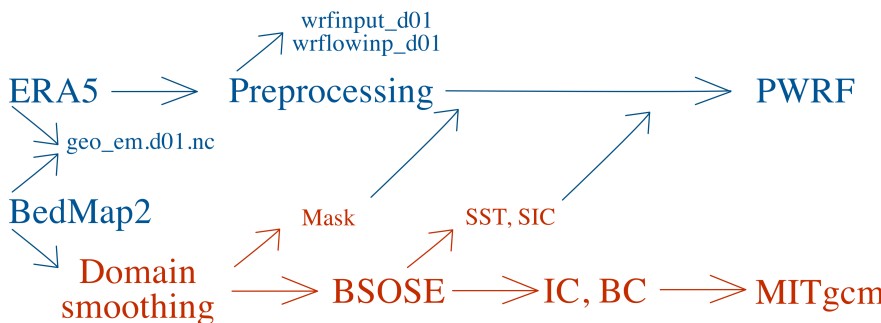

**Figure 2.** Workflow chart of the pre-processing and setting up of the experiments. geo_em.d01.nc, wrfinput_d01 and wrflowinp_d01 are the intermediate files generated by the PWRF pre-processing, IC and BC indicate initial conditions and boundary conditions, respectively.

(REMA, (Howat et al., 2019)) for the land topography and the land use product AntarcticaLC2000 (Hui et al., 2017), processed to be ingested by the WRF model by Gerber and Lehning (2020). The boundary conditions are updated every 6 hours using the ERA5 reanalysis (Hersbach et al., 2020). The physics options are the WRF single-moment 5-class microphysics scheme, the latest RRTMG option for both long and short wave radiation schemes (Iacono et al., 2008), the conjunction of the Janjic Eta Monin-Obukhov surface layer scheme and the Mellor-Yamada-Janjic boundary layer scheme (Janjic, 2002) and uses spectral
nudging above the 20th model layer for the u and v component of the wind, temperature and geopotential.

The preprocessing of the input files was conducted as is ordinary the case, but once the initialization step is performed, the input files are altered to match the coastline from MITgcm and the sea surface temperature and sea ice mask are updated to match those of MITgcm.

## 2.3   Model coupling

We base our coupled model on the SKRIPS model, described in Cerovečki et al. (2022) for the Southern Ocean. This model couples the MITgcm ocean component to the PWRF atmosphere part using the Earth System Modeling Framework (ESMF, (Hill et al., 2004)) according to the National United Operational Prediction Capability (NUOPC, (Sitz et al., 2017)) protocols and is available at https://library.ucsd.edu/dc/collection/bb1847661c and https://github.com/iurnus/scripps_kaust_model/. The study domain (Figure 1) extends from Victoria Land and Cape Adare (160°E) to Mary Byrd Land (140° W) along the coastline,
and from 64° south, encompassing the Ross Sea, to 88 ° south, including the whole Ross Ice Shelf and the Transantarctic Mountains. This domain has been designed to retain enough of the continent to allow katabatic winds and mesocyclones to develop (Carrasco and Bromwich, 1993; Carrasco et al., 2003) . The same polar stereographic 10 km grid is used for both models, allowing direct transfer of information between the oceanic and atmospheric components (and vice-versa) without the need of an additional step of regridding (although ESMF allows for multiple regridding options). Figure 2 illustrates the
workflow.



**Table 1.** Import of heat and mass fluxes versus calculations in the two experiments

|  |  | SKRIPS | | P-SKRIPS | |
|---|---|---|---|---|---|
|  |  | PWRF | MITgcm | PWRF | MITgcm |
| HEAT | latent heat | calc. | calc. | calc. | imp. |
|  | sensible heat | calc. | calc. | calc. | imp. |
|  | short wave net | calc. | calc. | calc. | imp. |
|  | long wave net | calc. | calc. | calc. | imp. |
| MASS | evaporation | calc. | imp. | calc. | imp. |
|  | precipitation | calc. | imp. | calc. | imp.** |
|  | sea ice runoff | calc. | calc. | calc. | imp. |
|  | land runoff | calc.* | N/A. | calc. | imp. |
| SNOW | | calc. | calc. | calc. | imp. |

calc. = calculated by the model

imp. = imported from PWRF to MITgcm

\* = calculated but is not accounted for (disappears)

\*\* = no snow deposition in *sea ice growth* routine

N/A = not applicable

## 2.4 Experiments

We run the coupled model in concurrent mode, with a 20 second timestep for the atmosphere component, and a one minute timestep for the ocean component. We perform two sets of simulations: one set for the summer month of January 2016 and one set for the winter month of August 2016. The experiments were set with the aim of testing the model skill for differ-

ent sea ice cover conditions (winter versus summer). There are 2 experiments for each of the sets, 1) the SKRIPS model reflecting the published set up of Cerovečki et al. (2022) and 2) our cast study including the conservation of heat and mass, the Polar-SKRIPS model version 1 (P-SKRIPS hereafter, Table 1). The code and experiments for this paper can be found at https://doi.org/10.5281/zenodo.7297744. Each of the simulations (1 and 2) is performed first over the whole month with a ten minute coupling timestep to evaluate performance, and secondly over only an hour with a one minute exchange timestep

and one minute output frequency to directly compare the exchanged variables at specific timesteps. For each of the runs, we initialise MITgcm sea ice concentration, sea ice thickness, and snow on ice thickness from the B-SOSE dataset, which then evolves freely within the MITgcm model (except for the snow on sea ice). The updated sea ice concentration mask is then passed on to PWRF.

### 2.4.1 SKRIPS case

The SKRIPS case reflects the set up for the Southern Ocean by Cerovečki et al. (2022) on which this study is based. Within this framework, the 2m temperature, specific humidity, 10m u and v components of wind, evaporation and precipitation rates,



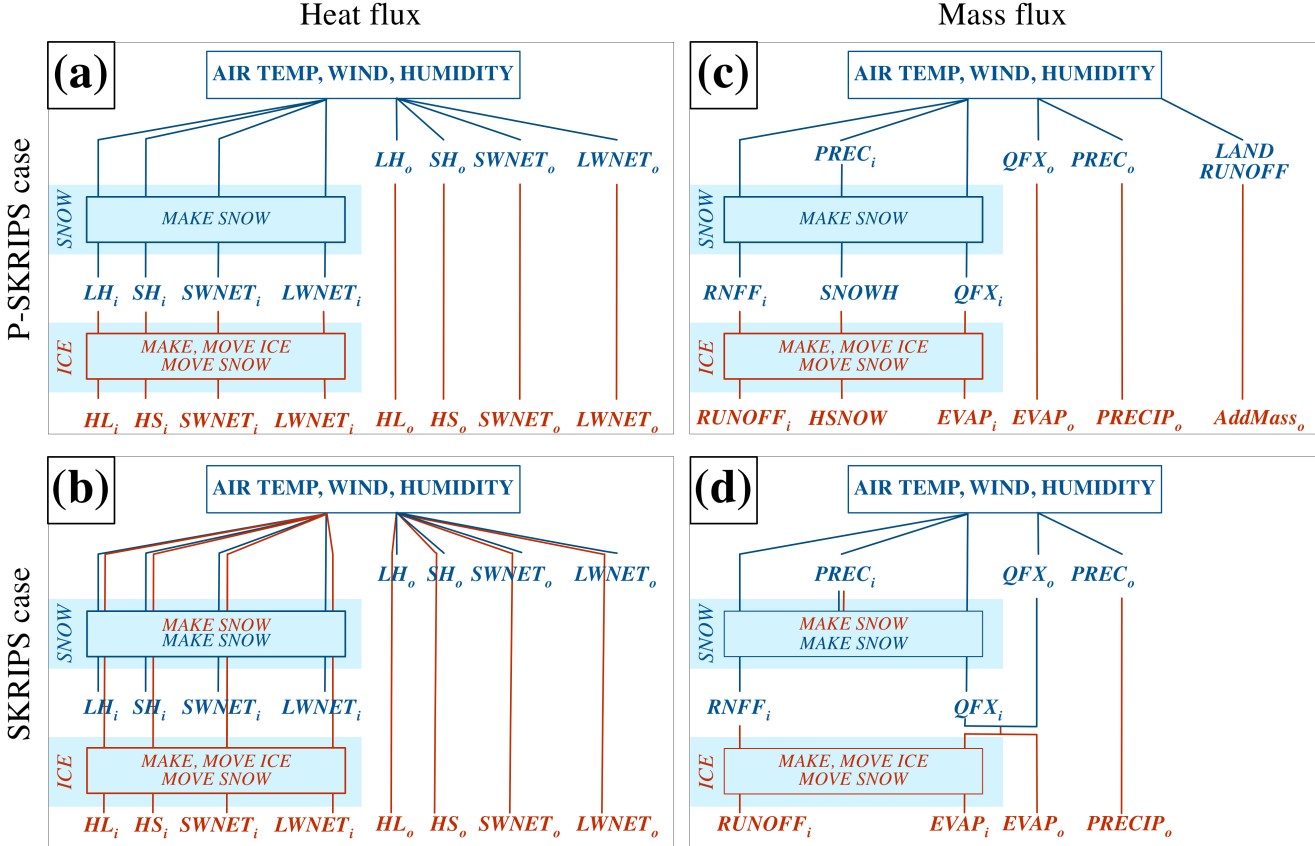

**Figure 3.** Exchanged flux pathways through snow and sea ice routines in PWRF (blue) and MITgcm (red), (a) and (c) represent the SKRIPS set up and (b) and (d) the P-SKRIPS. The $_i$ and $_o$ indices represent the variable over ice and ocean, respectively. LH, SH, SWNET and LWNET stand for latent heat, sensible heat, net shortwave radiation and net long wave radiation in W m$^{-2}$. Prec stands for precipitation, QFX and EVAP for surface evaporation and RNFF for surface meltwater runoff (all in mm). SNOWH is the variable for the amount of snow on sea ice (in m). In panels (b) and (d) the overlapping blue and red lines indicate the fluxes are re-calculated by MITgcm and WRF.

as well as downwards longwave and shortwave radiation are sent from PWRF to the ESMF/NUOPC coupler (Table 1). In MITgcm, all flux components (heat and freshwater mass) and the wind stress are re-calculated using the standard bulk formulae, with the meteorological variables received from PWRF. After the sea ice routine, the net surface heat and mass fluxes are scaled in the mixed cells containing ocean and sea ice in the ocean component. This parallel calculation of fluxes in the models is shown in Figure 3 (b,d).

The ocean variables are sent to the atmosphere to replace the reanalysis forcing in PWRF: the sea surface temperature, the sea ice concentration and thickness, and the sea surface currents



### 2.4.2 P-SKRIPS case

In this study, we attempt to conserve the heat and mass fluxes sent from the atmosphere into the ocean by importing the fluxes computed by PWRF to MITgcm directly, bypassing the recalculation in the oceanic component (Table 1).

In the original code, part of the energy flux is changed by the PWRF snow pack calculations prior to sending the field to MITgcm. Because MITgcm also has a snow on sea ice routine, it calculates an additional change on snowpack physics. To avoid this double change, we capture the flux variables after they have gone through the snow on ice physics in PWRF (just after the

seaice_noah call in surface_driver.F90) to pass to MITgcm. In addition, the calculation of fluxes (in seaice_solve4temp.F) and calculations of snow growth / melt (in seaice_growth.F) are turned off in MITgcm. PWRF calculates the surface evaporation and runoff resulting from the snow on sea ice melt. These freshwater fluxes are captured and transferred to MITgcm in the same fashion, including turning off the calculations representing these processes (in seaice_growth.F) in MITgcm. Precipitation over sea ice has been converted into sea ice surface snow thickness change by PWRF, we therefore do not account for snowfall

deposition in MITgcm. Finally, the land surface runoff is integrated for the whole ice shelf in PWRF and added in MITgcm at the coastline as a freshwater mass flux into the ocean (utilising the AddMass option in MITgcm). The schematic representation of this flow of fluxes is shown in Figure 3 (a,c). The coupling interface between models is defined as air-ocean over open ocean, between the sea ice and the snow on top of it, and below the snowpack on top of the ice sheet.

Since multi-layer snowpacks have been reported as an improvement for snow on sea ice representation (Arduini et al., 2022),

the Noah-MP multilayer snowpack is better suited to simulate the snowpack changes and properties in PWRF rather than snow on sea ice physics in MITgcm. The snow compaction/accumulation and thermodynamic changes (and interactions with radiative fluxes) all occur in PWRF, which then sends the snow thickness to MITgcm. The latter updates its snow on sea ice thickness variable accordingly. The ocean model then advects the sea ice (including the snow on top of it) and the new sea ice concentration and thickness are sent back to PWRF.

## 175 3 Results

We now present results of the different experiments, looking first at the conservation of heat fluxes. We describe how the sea ice mask affects each of the flux components exchanged by the P-SKRIPS model. Then, we look into the conservation of mass. We introduce the land runoff variable that describes the melting of snowpack on the ice sheet and its flow into the ocean. Lastly, we compare the models performance over a month-long simulation.


### 3.1 Conservation of heat

### 3.1.1 Import of the heat flux in the P-SKRIPS model

Figure 4 shows each of the fluxes exchanged or calculated in PWRF at 40 min, and MITgcm at 41 min (from the high frequency exchange simulation, the time step was chosen close to the start of the simulation). The one minute offset represents the fact





**Figure 4.** Maps of fluxes calculated or imported between models (PWRF, first column; MITgcm in the P-SKRIPS model, second column; MITgcm in the SKRIPS case, third column and difference between the MITgcm in P-SKRIPS and in SKRIPS in the fourth column). See Section 3.1.1 for model details. Plots for the January experiment; see August experiment in the Figure S1. Positive heat flux is defined upward.

that the fluxes are calculated in PWRF at 40 min (first column in Figure 4), and then imported into MITgcm at the 41th min. In the P-SKRIPS setup (second column in Figure 4), MITgcm directly receives the fluxes calculated in PWRF at the preceding time step, while in the SKRIPS case, MITgcm at 41 min uses PWRF 40 min output to recalculate the fluxes (third column in Figure 4). In order to differentiate the fluxes over sea ice from over the ocean surface, we implement additional diagnostic





variables in MITgcm (in seaice_solve4temp.F). The second and third columns in Figure 4 merge the MITgcm sea ice and ocean

components of each of the heat flux variables to ensure that we compare the same variables as in PWRF output.

The comparison of the first column (PWRF) and the second column (MITgcm in P-SKRIPS) indicate that the fluxes were successfully imported from PWRF to MITgcm. While the P-SKRIPS and the SKRIPS cases (columns 2 and 3) show similar flux distributions and magnitude of fluxes, there are visible differences between the two. The differences are not clearly attributable to any single variable, as all of the surface atmosphere and sea ice state parameters are included in the flux calculations. The

sea ice mask, the state of snow on ice (thickness, wetness, albedo), the calculation of surface temperature, and surface friction velocity in bulk formulas create a mosaic of differences between the heat flux maps. In January, the flux distribution differences are related to the presence or absence of sea ice. We attribute those differences to the calculation of fluxes in the sea ice package in MItgcm (maps in the last column of Figure 4). In contrast, most of the domain is covered by sea ice in August. At that time, longwave and shortwave net heat fluxes show more homogenous differences, reflecting different coefficients. The turbulent

heat fluxes are shown to be connected to the extra moisture brought by a precipitation event (Figure S1).

In general, the differences between the two cases reflect inconsistencies in the calculations between the two models: (1) there are differences in the momentum calculations used in bulk formulae for the latent and sensible heat components calculation, and the evaporation over snow and ice calculation; (2) small distinctions arise due to slightly different emissivity coefficients used in the calculation of the upward long wave radiation; (3) the differences in the upward shortwave radiation are a result of

conflicting abledo calculations between the two models, based on differing assumptions of dry/wet snow and snow threshold; (4) the sea ice mask varies between the two cases and PWRF internal heat fluxes calculations assess the presence of sea ice based on a predetermined threshold.

While there is no certainty that the PWRF calculations of fluxes perfectly represent reality, the PWRF snowpack is used rather than the MITgcm seaice fluxes calculation. PWRF's more sophisticated snowpack, accounting for more snow layers than the

MITgcm counterpart, is deemed to offer a better representation of snow on sea ice and improved boundary layer simulation (Arduini et al., 2022). By directly importing the heat fluxes from the atmosphere to the ocean and vice-versa, we avoid the differences induced by the parallel calculations of fluxes in the two components that would lead to inconsistencies between the flux in the two coupled models. We thus highly recommend the import of fluxes in each model component in order to conserve the fluxes for the general balance of the coupled model.

**3.1.2   Scaling by the sea ice concentration mask**

We initially import the flux components from PWRF into MITgcm using the variables available from the PWRF output files. The import of surface fluxes in the exf package is prohibited when using the sea ice package in a standard version of MITgcm, because the fluxes need to be calculated according to the sea ice mask. In the coupled set up, the sea ice mask is coordinated between PWRF and MITgcm, so the fluxes can be directly used in the sea ice packages, without any mask mismatch. We thus

remove this error flag from seaice_model.F (in MITgcm). This is shown as Direct Import in Figure 5 (b), where the path of a variable X is shown. X refers to any component of the heat flux: latent, sensible, shortwave and longwave next fluxes. In the Direct import, $X_{PWRF}$ is imported as is into the MITgcm exf package. However, the flux components have already been



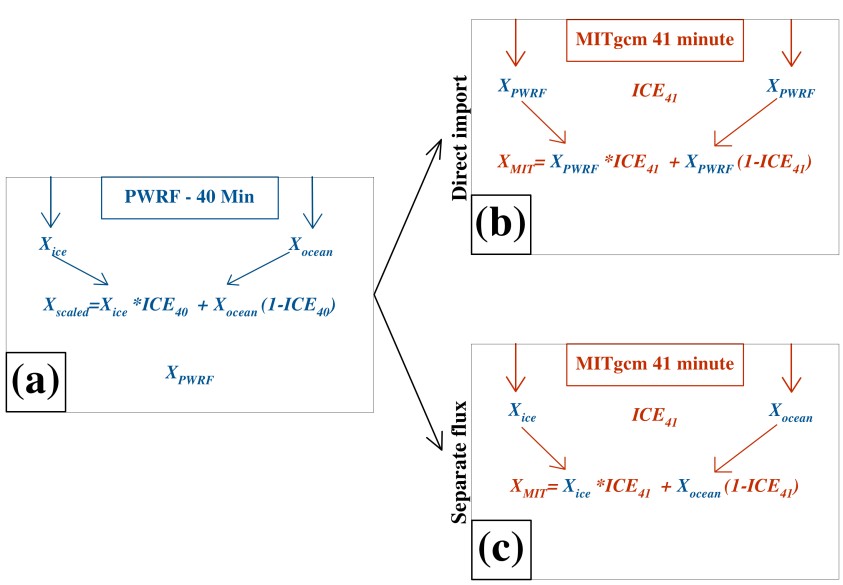

**Figure 5.** (c) Separate fluxes variables exchange versus (b) direct import of the PWRF output (a) approach. X represents any heat flux component in both models. $X_{ice}$ is the flux value over sea ice, $X_{ocean}$ is the flux value over the open ocean. $X_{scaled}$ is calculated mid-timestep in PWRF, while $X_{PWRF}$ is available as output at the end of the timestep. $ICE_{40}$ and $ICE_{41}$ represent the sea ice concentration calculated by MITgcm at the respective timestep. $X_{MIT}$ is the heat flux received from PWRF and available for the ocean model after scaling (not calculated for each flux component in MITgcm, only for the total heat flux, but used here to directly compare to $X_{WRF}$).

scaled in PWRF to account for mixed grid cells, partially occupied by sea ice. This occurs mid-timestep (surface_driver.f90 in PWRF, Figure 5 (a), $X_{scaled}$) and the model output contains these scaled variables. This implies that when MITgcm receives

the variable and accounts for the presence of sea ice in its own routines, the heat fluxes are scaled twice, thereby affecting the amount of heat gained or spent over the sea ice areas. To tackle this issue, the separate heat flux components are captured in PWRF over the sea ice and over the ocean separately (5 (a), $X_{ice}$ and $X_{ocean}$). In the Separate fluxes approach two variables are now imported for each heat flux component: $X_{ocean}$ becomes the exf variable; and $X_{ice}$ is used instead of the recalculation of fluxes (solve4temp.F in MITgcm) to determine the amount of heat available for sea ice growth/melt (F_ia variable).


Figure 6 shows an example of this approach for the latent heat flux: (a) $X_{PWRF}$ , (b) $X_{MIT}$ , (c) $X_{PWRF}$ - $X_{MIT}$ and (d) sea ice concentration difference $ICE_{40}$ - $ICE_{41}$. The Separate fluxes approach reduces the differences between the two models to negligible amounts (Figure 6(c)). Firstly, the differences between $X_{scaled}$ and $X_{PWRF}$ (Figure S2 (a)) are due to the fact that the heat fluxes variables are affected by subsequent processes after the mid-timestep scaling in PWRF, when these variables are

captured to be sent to MITgcm. Such processes encompass albedo changes affecting the upwelling short wave, the skin surface temperature and the upwelling long wave radiation, but these are trivial compared to the range of the exchanged fluxes. Due to



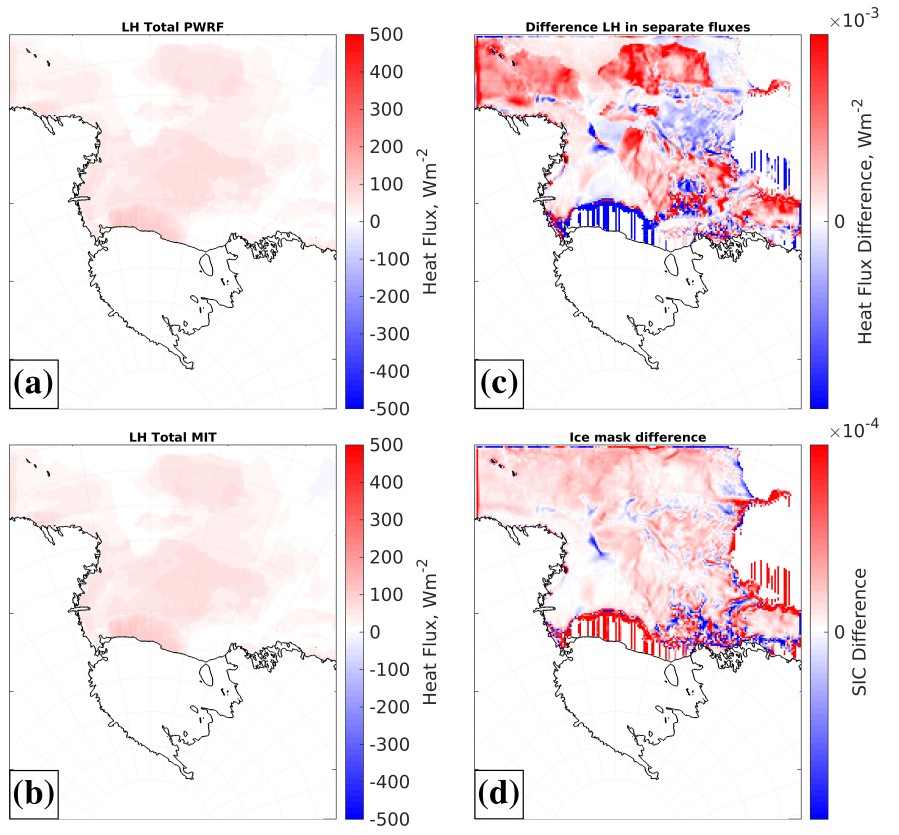

**Figure 6.** (a) Latent Heat (LH) flux: total heat flux in PWRF using the capture of the Separate fluxes, (b) latent heat in MITgcm, (c) differences between (a) and (b) and (d) ice concentration mask difference between the two timesteps. See Section 3.1.2 for more details.

the nature of the coupling, the sea ice concentration mask evolves between minutes 40 and 41 (Figure 6 (d)). Thus, the values of $X_{PWRF}$ correspond to the flux using the sea ice concentration sent from MITgcm at 40 minutes. PWRF sends $X_{PWRF}$ to MITgcm to be used for timestep 41, using the sea ice concentration mask at that timestep. Hence even in the Separate Fluxes approach, $X_{scaled}$ and $X_{MIT}$ are not exactly the same. Because the ice concentration mask difference is small, the scaling multiplication by $X_{ICE}$, which is  104 (Figure 6 (d)), results in subtle differences in the order of $10^{-3}$ W m$^{-2}$. The sign definition of each of the flux components is listed in Table S1. We note that additional heat is spent on sea ice thermodynamics by MITgcm (in seaice_growth.F), so the resulting sum of fluxes available to the ocean model (Qnet) is not the same as a sum of the four heat flux components ($X_{MIT}$). This is why we created extra diagnostic variables.





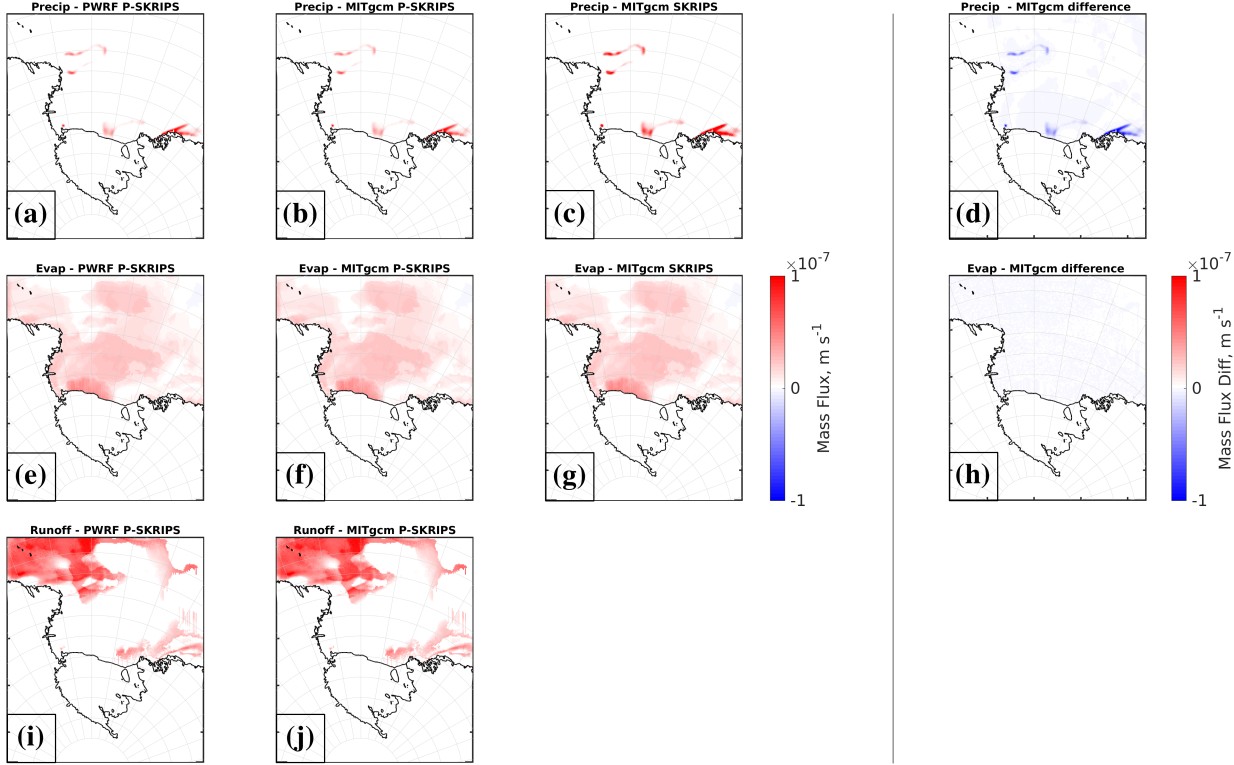

**Figure 7.** Maps of fluxes calculated or imported between models (PWRF, first column; MITgcm in the P-SKRIPS model, second column; MITgcm in the SKRIPS case, third column and difference between the MITgcm in P-SKRIPS and in SKRIPS in the fourth column). See Section 3.1.1 for model details. Plots for the January experiment; see August experiment in the Figure S3. Positive evaporation is defined upward, positive precipitation and runoff are defined downward.

## 3.2 Conservation of Mass in the P-SKRIPS model

### 3.2.1 Precipitation, evaporation and runoff

To avoid repeated accumulation of snowfall over sea ice in each model component, the deposition of snow is neglected in MITgcm (seaice_growth). The new mass of snow is accounted for by the exchange of the height of the snow on sea ice variable (SNOWH) sent by PWRF to MITgcm after the deposition, compaction and metamorphism of the snow cover have taken place in the atmospheric component. The snow falling over the open ocean, however, is taken care of by MITgcm.

The evaporation at the snow/ice surface due to interactions with the atmosphere is another process that is accounted for in both PWRF and MITgcm. Analogous to the heat fluxes, in the coupled model, the snowpack module of PWRF generates the





evaporation flux that is directly imported by MITgcm (in seaice_growth). In addition, this time the scaling to resolve mixed ocean-sea ice cells is ignored in MITgcm (in seaice_growth) as it was already applied in PWRF. Figure 7 shows a similar range

and pattern for the two simulations, where the variations occur for regions of differing sea ice concentration mask.

Finally, the melt occurring at the surface of the snow and not retained in the snowpack (by refreezing) is exported in the form of runoff in PWRF (SFCRUNOFF). In MITgcm, the input of freshwater from runoff usually comes from river runoff but is used here to collect the water melted from the snow layer present over the sea ice. This variable is directly imported from PWRF and is captured in the atmosphere component by adding an option to store the variable over sea ice (in addition to the land-only

variable). This evolution is also noticeable in Figure 7 as the differences between columns 2 and 3 reflect the varying amounts of surface meltwater calculated by each of the models independently (SKRIPS case), or directly imported into MITgcm from PWRF (P-SKRIPS case).

### 3.2.2 Ice sheet runoff

In addition to the meltwater runoff over sea ice, PWRF generates another surface mass balance term over land: runoff from the

surface of the ice sheet that needs to be redirected to the ocean. In the standalone simulations of PWRF, the meltwater that does not refreeze into the snowpack and runs off is treated as a sink term and does not affect any other component. In the coupled framework, we identify the ice sheet meltwater runoff as an important component of the total mass budget that has an effect on any future projections (Golledge et al., 2019). In that regard, the land ice meltwater runoff is captured in PWRF. Because the land surface runoff is a cumulative variable, only the change in surface runoff (i.e. the amount from that timestep minus the

amount from the previous timestep) is passed to the coupler and to MITgcm. The land surface runoff is integrated along the y-axis (meridional integration) within the ESMF-MITgcm interface (get_field_parameters.F90, a new variable, land_runoff, is added as a part of the exf package).

We use the MITgcm AddMass variable to add this meltwater into the ocean component. First, we create a mask variable (AddMassInit), read from an external file, and fixed in time. It is used to mask the integrated land_runoff from the atmosphere

component. This creates a variable AddMass null everywhere, except along the front of the Ross Ice Shelf, where it has the integrated value of the land surface runoff to be added to the ocean as freshwater.

However, this approach creates issues. First, the meltwater is not collected from the whole of the ice sheet within the domain, but only from cells at the same x-axis value as our mask identifies, which is along the coast of the ice shelf in an east-west direction. Thus, we are missing meltwater runoff from the Drygalsky Ice Tongue, from other small ice shelves, and from the

eastern and western edges of the Ross Ice Shelf that are not aligned with the ice shelf front (not at the same x-axis value in the grid). Second, we assume no lag for the runoff getting into the ocean, implying that the meltwater runoff originating at the southernmost part of the domain is added to the ocean within one coupling timestep (one minute), not accounting for delay in the time it takes the meltwater to arrive to the ice shelf front. This will be addressed in the future developments.

The code has been tested with a test value in the coupler (not shown), but the January and August 2016 simulations do not

generate any meltwater runoff over the ice sheet due to the limited simulation time, meaning that the import of the variable





gives a net zero flux. This new surface mass balance variable is nevertheless added to the coupled model in order to close the mass balance in future work.

### 3.3 Flux differences during a one month experiment

Figure 8 displays the January 2016 time series for each of the integrated fluxes presented in Figure 4 and Figure 7. We integrate
all flux components over the ocean (we ignore land), and present the fluxes at the atmosphere-ocean or snow-sea ice interface, following Figure 4 (through the coupling interface). The PWRF P-SKRIPS and MITgcm P-SKRIPS curves overlap perfectly for each of the variables; they are identical. There are small differences in the PWRF flux variables between the P-SKRIPS and the SKRIPS cases, expressed either as larger minima (in the case of short wave fluxes) or as divergence over time. This is because the simulations show different responses to the conservation of fluxes versus parallel calculation. In summer, the in-
coming shortwave radiation is at its maximum in the middle of the day and the differences in albedo parameterization between PWRF and MITgcm account for the variations in the magnitude of the shortwave peak.

The largest discrepancies in heat fluxes are between the MITgcm results from the direct import method used in P-SKRIPS case and the independent calculation used in the SKRIPS case. In summer, differences in the shortwave net calculations be-
tween PWRF and MITgcm in the SKRIPS case dominate the heat flux inconsistencies (on average, an order of magnitude larger than the turbulent heat fluxes differences, Table S2). In August, when the sun is low on the horizon, the sensible heat and longwave radiation fluxes show the largest differences between PWRF and MITgcm. At midday when solar radiation is available, small amounts of snow on sea ice melt and their impact appears in the evaporation, the long wave net, and the sensible and latent heat fluxes, in the form of repeating, short lived peaks (Figure S4).
The ocean receives a larger amount of latent heat in the SKRIPS simulation with an almost constant bias of 1.1013 W. The differences in sensible heat flux into the ocean varies over time between the two models. Larger discrepancies are associated with precipitation events (6th-8th of August, 12th of August, 14th-19th of August, and 26th-31th of August, with 29th excepted), leading to a loss of heat from the ocean for the SKRIPS simulation, while the P-SKRIPS simulation indicates a gain for the ocean. The longwave net radiation in the SKRIPS case is always slightly underestimated, as compared to the P-SKRIPS, 
although both curves show very similar variations over time.

Despite the fact that precipitation is directly imported from PWRF to MITgcm (also in the SKRIPS case), the amount of precipitation sent by PWRF to MITgcm is overestimated. This is because in the SKRIPS case the total precipitation is made out of the sum of the time step non-convective precipitation (RAINNCV) and non-convective snow and ice (SNOWNCV). However, the latter is defined as a component of the non-convective precipitation, which encompasses all species (rain, graupel and snow and ice) and, therefore, is accounted for twice in the snowfall term. In the P-SKRIPS, the components are added individually 
and the time step non convective precipitation term is ignored.

As evaporation is directly imported from PWRF to MITgcm in both cases, no difference exists between the two simulations. In the last week of the simulation, the fluxes in the two experiments diverge more and more, due to the (now correct) balancing of the fluxes.



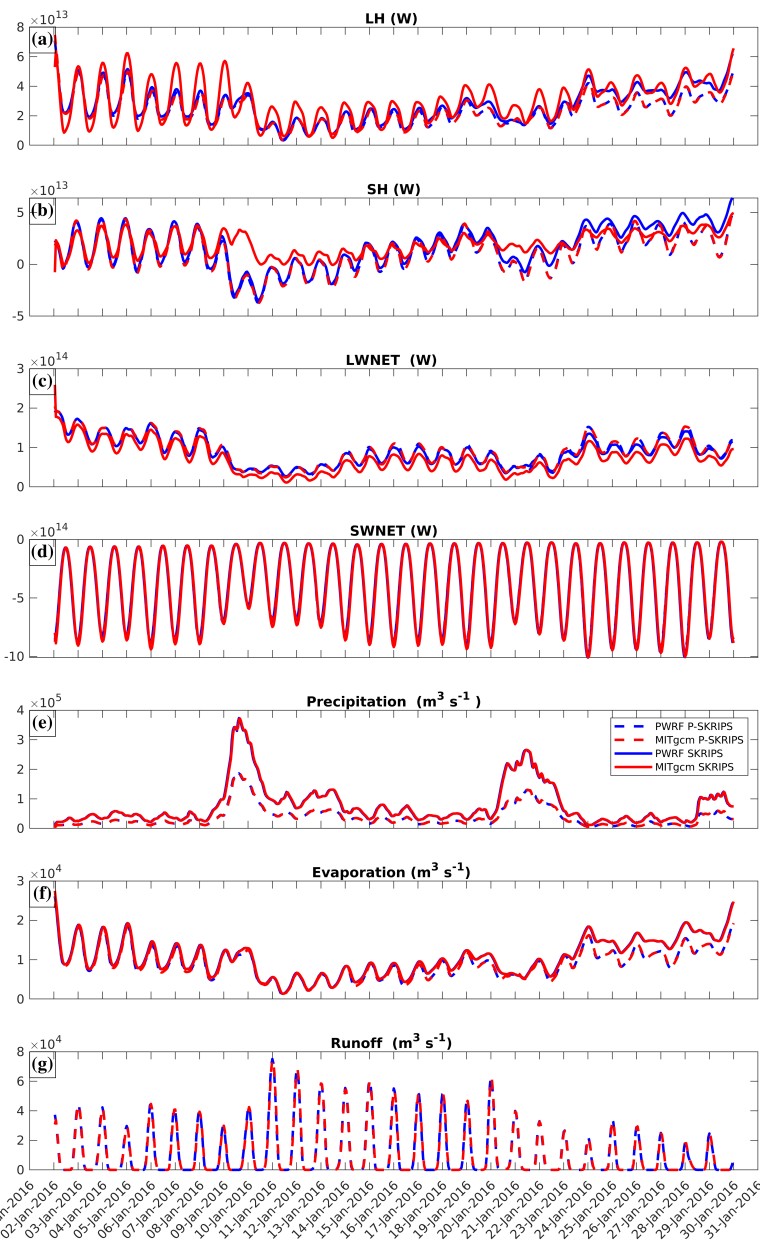

**Figure 8.** Integrated flux time series through the coupling interface for the January 2016 experiment. P-SKRIPS case is dashed, SKRIPS case is in solid lines. PWRF fluxes are in blue, MITgcm are in red. Heat fluxes are defined positive upwards, the evaporation is defined positive upwards, the precipitation and runoff are defined positive downwards. For more details see Section 3.3. For the August experiment results, see Figure S4.



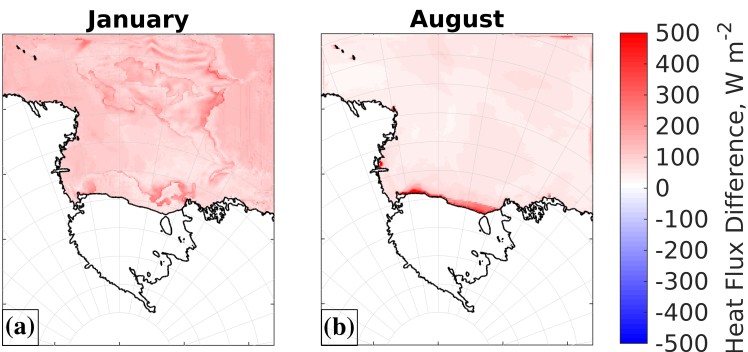

**Figure 9.** Maximum differences of total heat flux exchanged for the January and August 2016 experiments.

Finally, the runoff over sea ice is an additional modification in the P-SKRIPS that does not exist in the SKRIPS case. Therefore, only the PWRF and MITgcm balanced outputs are present, and match perfectly.

The differences in heat fluxes calculation between PWRF and MITgcm have a clear spatial distribution, and the highest discrepancies are linked to areas of open water along the coast in August (ocean polynyas). Differences in the Figure 9 and in the text hereafter are defined as PWRF-MITgcm, with heat fluxes defined as positive upward. Due to the strong winds keeping

the polynyas open, the turbulent heat fluxes differ significantly in the SKRIPS simulation: maximum differences are 922 W m$^{-2}$ for sensible heat flux and 318 W m$^{-2}$ for latent heat flux (PWRF producing larger values). The total heat flux difference ranges from -295 to 1212 W m$^{-2}$ in August. In January, the difference of the total heat flux is smaller (-466 to 392 W m$^2$) and is comparable with the shortwave heat flux difference alone (-455 to 424 W m$^{-2}$). Maximum absolute differences are located at the edge of the sea ice, where snow on ice calculation is most important for the total heat flux. These reflect the imbalance of

heat flux calculations in coupled models, now corrected in the P-SKRIPS model. It highlights the importance of conservation of heat and mass fluxes, especially when polynyas are considered.

## 4 Discussion and conclusion

In this paper we have conserved water and energy fluxes between the ocean and atmosphere components of the P-SKRIPS coupled model set up for the Ross Sea domain. We have provided a detailed analysis of flux pathways in each of the components

of the coupled model and described strategies for the conservation of the variables exchanged. One-month test experiments for the January and August of 2016 have been carried out and show that the P-SKRIPS approach homogenises the ocean and atmosphere calculations. This is an improvement and is needed for balanced long term simulations, especially in coastal polynyas.

The representation of sea ice is split between the ocean and the atmosphere components. The formation and advection of

sea ice stay in MITgcm. Snow cover changes happen in PWRF. Its multi-layered snowpack and complex representation of



snow on sea ice makes PWRF the better model to represent the snow/ice-atmosphere interactions. As stated in Arduini et al. (2022): firstly, by its insulating effect, reducing the heat exchange from the ocean through the sea ice and snow on sea ice to the atmosphere, enabling the representation of strong cooling events and surface based inversions. Secondly, by having better (several layers of snow) schemes, changes in the atmosphere are better captured and the snow responds quicker to those changes. Thirdly, the ratio of snow on sea ice and sea ice thickness is thought to be almost constant in Antarctica (Jeffries et al., 1998; Toyota et al., 2011; Worby et al., 1996), which implies that changes in thickness of the snow on sea ice impact the sea ice thickness (Toyota et al., 2016). In addition, the snow on sea ice initialisation depends on a couple of settings in the atmospheric component, and prescribing a too thin snow cover (0.0005 m, seaice_albedo_opt = 1 in PWRF namelist.input) leads to a unrealistically large negative latent heat flux over sea ice ( 400 W m$^{-2}$). Hopefully, more detailed and accurate observations of the snow on sea ice cover will be available in the future, to help the initialisation of this parameter in the models, and to enable their validation.

We identify a series of potential improvements to P-SKRIPS that are listed below, and will be the focus of future research:

- In summer, large differences in shortwave fluxes are directly linked to the calculation of the albedo, which varies between the model components as they have different parameterization of dry/wet snow. We believe that exchanging the snow on sea ice albedo value between the models, as suggested by Ren et al. (2021) would be an improvement but it requires a series of alterations in the sea ice routine in PWRF (module_sf_noah_seaice.F90) to facilitate the use of the albedo variable.

- The two-way exchange of snow on sea ice variables between the atmosphere and the ocean models would allow the snow cover thermodynamic changes to take place in PWRF. This would send the snow on sea ice thickness information to MITgcm, that could then advect and flood the snow on top of sea ice as the ocean moves it around. At the moment, the snow on sea ice thickness information is passed from PWRF to MITgcm, moved around by the ocean component, but not sent back to PWRF.

- Sending back the snow on sea ice variable to PWRF once it has been advected in MITgcm, is an important future development. Currently, PWRF does not have the ability to remove the snow on ice once the sea ice has been advected: if a sea ice cell moves and is replaced by an open ocean cell, the snow on sea ice layer will remain present in the now ocean cell. The snowpack calculations occur only for sea ice-filled cells, thus PWRF ignores the snow on sea ice values for now open ocean cells. The "old" snow on ice will be used in case of new sea ice advection or growth at this location, e.g. within the polynyas where sea ice is constantly moving. This should be addressed in the future in order to get rid of the relict of the snow on sea ice layer over ocean cells.

- The momentum variables have not been balanced in this version of the coupled model: the wind stress is calculated and accounted for in MITgcm, using the u and v wind components passed on from the atmosphere.

– Bulk fluxes calculated by the ocean and atmosphere are quite different. The sea ice package in MITgcm was created for and tuned according to the ocean model. It needs further refinements to work smoothly with the directly imported fluxes from PWRF. In short, in the balanced coupled model, MITgcm needs tuning to the new atmospheric forcing.

The development of the coupled Polar Skripps-KAUST PWRF-MITgcm model presented in this paper makes conservation of heat and mass possible over long term simulations, which is essential for more reliable future projections. The non-conservation of up to 922 W m$^{-2}$ is directly affecting the heat content of the atmosphere and deep convection of the ocean. While the atmosphere's response the extra energy is the formation of localized weather events, the ocean's response to a reduced amount of heat flux leads to a weakening of the formation of the Antarctic Bottom Water in polynyas in the Ross Sea.

The presented coupled model setup constitutes, to our knowledge, the most accurate representation of ocean/atmosphere/sea ice interactions for polar climates and is thus recommended for climate modelling in any Arctic and Antarctic region.

*Code and data availability.* The developments and files required to set up and run the model presented in this paper are available at https://github.com/alena-malyarenko/P-SKRIPS and on Zenodo https://doi.org/10.5281/zenodo.7297744. The coupled model builds on the Scripps-Kaust model described in Cerovečki et al. (2022) and Sun et al. (2019). The base code can be find at

https://github.com/iurnus/scripps_kaust_model/. ERA5 Reanalysis (0.25 Degree Latitude-Longitude Grid), generated by European Centre for Medium-Range Weather Forecasts are available at https://cds.climate.copernicus.eu/cdsapp#!/dataset/reanalysis-era5-pressure-levels and https://cds.climate.copernicus.eu/cdsapp#!/dataset/reanalysis-era5-single-levels. BSOSE data is availbe at http://sose.ucsd.edu/. The Bedmap 2 data can be downloaded from https://www.bas.ac.uk/project/bedmap-2/.

*Author contributions.* AM and AG designed the study and developed the P-SKRIPS version of the model. MK and RS helped with the

technical implementation of the model developments. Visualisation and analysis of results have been carried out by AM and AG. All authors contributed to writing and editing the manuscript.

*Competing interests.* The authors declare no competing interests.

*Acknowledgements.* AM, AG and MK are funded by the New Zealand MBIE-funded Antarctic Science Platform (ANTA 1801). The modelling has been supported by the New Zealand e-Science Infrastructure (NeSI). AM and AG would like to thanks Alexander Pletzer, Chris

Scott, Wes Harrell and Aleksandr Beliaev for technical help.



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
