# Peer review of "Conservation of heat and mass in P-SKRIPS version 1: the coupled atmosphere-ice-ocean model of The Ross Sea"

_EGUsphere, 2022_

## Author Comment (AC3)

Author response to the Referee Comments by Anonymous Reviewer 1 on the manuscript:

**Conservation of heat and mass in P-SKRIPS version 1: the coupled atmosphere-ice-ocean model of The Ross Sea**

Alena Malyarenko, Alexandra Gossart, Rui Sun, and Mario Krapp

submitted to Geoscientific Model Developments (https://doi.org/10.5194/egusphere-2022-1135)

We thank the Reviewer for all the time and effort put into the review of our manuscript and are pleased with their positive and constructive comments. Please find the response to each of the comments below. The reviewer's comments are displayed in **bold text**, replies are shown in normal text, text from the original manuscript is shown in blue, and proposed changes to the manuscript are shown in red.
* * *
**Review of Conservation of heat and mass in P-SKRIPS version 1: the coupled atmosphere-ice-ocean model of The Ross Sea" by Alena Malyarenko, Alexandra Gossart, Rui Sun and Mario Krapp.**

**Summary: The authors present a new version of a coupled regional model of the Ross Sea region in which the coupling between the atmosphere and ocean components now conserves heat and mass, in contrast to previous versions of such models. Overall I find this to be a very valuable contribution to the modelling of polar regions and recommend publication after addressing the relatively minor comments below.**

**1 General comments:**

**Overall, while the results presented by the authors are compelling, they are somewhat oversold. I have highlighted some specific instances below but in general the authors refer to their new version of the model as being "better" or "more accurate" with no (by design) quantification of what this means. The authors note that they do not spin up the model as they are not attempting to simulate the Ross Sea in this paper, but rather aim only to highlight the technical advancements of the model. I commend the authors for noting this explicitly, but it then makes it difficult to understand what they mean by "better". I recommend checking instances of such language (many of which I have highlighted below) and either clarifying what the authors mean by "better" or "more accurate", or rephrasing.**

We thank you for this comment, and will make sure the language has been checked to rephrase such instances.

**My second general comment is on organization. It is quite muddled where various parts of the technical changes the authors have made to the model are within the paper. Some are in the "Methods", some in the "Experiments" and some in the "Results" sections. In particular, I found it odd and difficult to understand why Section 3.1.2 "Scaling by the sea ice concentration mask" is in the results section. Surely the description of the contrasting methods by which this scaling is done should be in the Methods, and then only the results contrasting the two shown in the Results section? As I have noted below, I also found it difficult in this section to understand what the contribution of the authors is in this section. See below for details.**

We thank you for this comment, which is addressed in the minor comments below.

**2    Minor comments:**

**Line 18: Maybe "mass balance" would make more sense here than "mass gain"?**

Thank you, we will replace "mass gain" by "mass balance".

For example, the Southern Ocean is the main source of moisture for the Antarctic continent (van Wessem et al, 2018; Agosta et al, 2019) and it controls the mass gain of land-locked ice and thus global sea level (e.g. Holland et al., 2020; Golledge et al., 2015; Krinner et al., 2007).

For example, the Southern Ocean is the main source of moisture for the Antarctic continent (van Wessem et al., 2018; Agosta et al., 2019) and it controls the mass balance of land-locked ice and thus global sea level (e.g. Holland et al., 2020; Golledge et al., 2015; Krinner et al., 2007).

**Lines 30-36: Would be worth discussing in this section about the fact ESMs don't have coupled ice sheets or ice shelf cavities, but that regional models can.**

**Line 38-39: "their global physics are not optimised for polar areas". What does this mean? Can you provide specific examples? I also looked at the Agosta et al., 2015 paper that is cited but it also just makes this claim with no substantiation or reference. More detail needed.**

Thank you for the two remarks above. We decided to address them together. First, as described by Smith et al. (2021), GCM are typically run at low resolution and can not represent all the processes. For example, refined spatial patterns of accumulation and melt processes at the ice sheets - atmosphere/ocean interface can not be represented in GCMs. Which leads to static ice sheet boundaries and heavy parametrization, limiting the inclusion of ice sheet/ice shelves cavities models into GCMs. This implies that the physics of ice and the ice-ocean and ice-atmosphere interactions are usually not accurately represented in GCMs.

Secondly, the coarse resolution of GCM limits them in the representation of local scale and regional features (e.g., the Antarctic Peninsula) and the parametrization of physical processes occurring at finer resolution prevents the adequate representation of regional and local scale phenomena, especially in complex areas (Bozkurt et al., 2021).

ESMs that are part of the Coupled Model Intercomparison Project (CMIP) experiments, have coupled global ocean, atmosphere, land and sea ice models (Meehl et al., 1997), but their spatial resolution is rather coarse. In addition, their global physics are not optimised for polar areas (Agosta et al., 2015), leading to various performances in the Arctic and Antarctic. Although there is a general improvement since previous experiments, the CMIP6 (Eyring et al., 2016) models still represent an ocean surface that is too warm and too fresh for the Southern Ocean, with a too small annual sea ice extent (Beadling et al., 2020). The models struggle to represent realistic sea ice cover (Mohrmann et al., 2021) and show a wide spread in the different terms of sea ice formation/dissipation, leading to large uncertainties in the sea ice budget across the different models (Li et al., 2021; Roach et al., 2020).

ESMs that are part of the Coupled Model Intercomparison Project (CMIP) experiments, generally have coupled global ocean, atmosphere, land and sea ice models (Meehl et al., 1997). However, the global atmosphere and ocean models that make up ESMs are not optimized for polar areas (e.g. Hines et al., 2008, Azaneu et al., 2014) and polar versions of these models are developed to represent processes specific to these regions. In addition, the spatial resolution of ESMs is rather coarse, which prevents them from representing local or regional-scale processes. For example, Smith et al., (2021) raises the fact that accumulation and melt at the ice-ocean-atmosphere interface have refined spatial patterns that can not be represented in GCMs. And this leads to static ice boundaries and heavy parametrization of these processes, limiting the inclusion of refined ice sheet or ice shelf cavity models into GCMs. Therefore, ice-ocean and ice-atmosphere interactions are usually not accurately representend into GCMs.

In addition, the parametrization of processes occurring at higher resolution in GCMs physics limits them in the representation of local scale and regional features (e.g., the orography and associated local processes of the Antarctic Peninsula, Bozkurt et al., 2021), indicating that the global physics of GCMs are not optimised for polar areas (Agosta et al, 2015; Bozkurt et al., 2021), leading to various performances in the Arctic and Antarctic. Although there is a general improvement since previous experiments, the CMIP6 (Eyring et al., 2016) models still represent an ocean surface that is too warm and too fresh for the Southern Ocean, with a too small annual sea ice extent (Beadling et al., 2020). The models struggle to represent realistic sea ice cover (Mohrmann et al., 2021) and show a wide spread in the different terms of sea ice formation/dissipation, leading to large uncertainties in the sea ice budget across the different models (Li et al., 2021; Roach et al., 2020).

**Line 57: Table reference is broken**

We thank you for highlighting this. This is an error as no table is planned to be inserted here, we will remove the reference to the table.

**Line 98: NCAR is the National Center for Atmospheric Research (American spelling of center and not plural).**

Thank you, we will change the text accordingly.

It is a non-hydrostatic mesoscale atmospheric model developed by the NCAR (National Centres for Atmospheric Research) for both atmospheric research and operational forecasting applications.

It is a non-hydrostatic mesoscale atmospheric model developed by the NCAR (National Center for Atmospheric Research) for

both atmospheric research and operational forecasting applications.

**Line 107: "and has a closer snow water equivalent reproduction" closer to what? What is it reproducing?**

The improvements made to the model led to a better representation of a series of snow surface properties and fluxes in the surface snow cover. One of them is a representation of the snow water equivalent (the amount of liquid water present in the snow) that is closer to the reality. We agree that the sentence is somewhat incomplete and propose to rephrase as follows:

It improves the representation of surface fluxes, surface meltwater production, percolation and retention/refreezing in the snow layers and surface runoff, and has a closer snow water equivalent reproduction with an improved diurnal cycle of the snow skin temperature.

It improves the representation of surface fluxes, surface meltwater production, percolation and retention/refreezing in the snow layers and surface runoff, and has a refined snow water equivalent reproduction (closer to reality) with an improved diurnal cycle of the snow skin temperature.

**Line 116: "as is ordinarily the case"**

Yes, we will correct this. In addition, Reviewer 2 asked to add some details, so the sentence will be changed:

The preprocessing of the input files was conducted as is ordinary the case,

The preprocessing of the WRF input files followed the standard procedure (ungrib.exe - metgrid.exe - real.exe, Wang et al., 2019),

**Section 2.3: "Model coupling" seems like the wrong title for this section. Maybe "Model domain" would be better?**

We agree and will rename the section title accordingly.

**Line 121-122: When combining an acronym with a reference in parentheses, the parentheses are not needed around the reference also. Can be done in latex with \cite{Hill2004} or \citep[ESMF][]{Hill2004} or similar.**

Thank you, this detail was overlooked in the first submission. We will change the citation when combining acronyms and references in the new version of the manuscript.

**Line 179: "we compare the models performance" against what benchmark? You mentioned earlier that you were not spinning up the model since you are not aiming to simulate the Ross Sea realistically, which is fine, but then you should mention here what you mean by performance.**

We agree that the wording was not ideal. We propose to repharase as follows:

Lastly, we compare the models performance over a month-long simulation.

Lastly, we compare the results of the different model set ups over a month-long simulation.

**Line 185: 41st not 41th**

Yes, this will be corrected.

and then imported into MITgcm at the 41th min.

and then imported into MITgcm at the 41st min.

**Line 199: Should this say ", reflecting different emissivity coefficients."? Unclear what coefficients this is talking about.**

Thank you, we will correct this in the new version of the manuscript.

At that time, longwave and shortwave net heat fluxes show more homogenous differences, reflecting different coefficients.

At that time, longwave and shortwave net heat fluxes show more homogenous differences, reflecting only different emissivity coefficients and albedo values.

**Line 205: Albedo spelled incorrectly**

Thank you, we will correct this in the new version of the manuscript.

the differences in the upward shortwave radiation are a result of conflicting abledo calculations between the two models,

the differences in the upward shortwave radiation are a result of conflicting albedo calculations between the two models,

**Section 3.1.2: I found this section confusing, as I am not sure what the result is here. Are the authors:**

 – **Contrasting two existing ways the model does this scaling and commenting on which is best? If so, make this clearer and explicitly say which is better and why.**

 – **Comparing the existing way of doing the scaling with a new method the authors developed? If so, make it clearer that you are saying the existing way is deficient in some way that you are fixing**

 – **Comparing two new options that the authors have developed? If so, as in option (a) make this clearer and explicitly say which is better and why.**

 **and**

**Figure 5. This way of describing what is shown in each panel was confusing to me. Would suggest to describe panels (a), (b) and (c) in order. Also, should $X_{WRF}$ at the end of the caption be $X_{PWRF}$?**

We decided to collate those two comments and address them at the same time.

We agree that the original section 3.1.2 can be more explicit. We will make sure this is stated clearly. In addition, and in response to you second general comment, we will move section 3.1.2. to 2.5. and adapt it accordingly.

We also agree that it would be more convenient to describe the panels in Figure 5 in order and propose to revise the paragraph as follows. And yes, it should be $X_{PWRF}$ and not $X_{WRF}$ at the end of the caption.

We initially directly import the flux components from PWRF into MITgcm using the variables available from the PWRF output files. The import of surface fluxes in the exf package is prohibited when using the sea ice package in a standard version

of MITgcm, because the fluxes need to be calculated according to the sea ice mask. In the coupled set up, the sea ice mask is coordinated between PWRF and MITgcm, so the fluxes can be directly used in the sea ice packages, without any mask mismatch. We thus remove this error flag from seaice_model.F (in MITgcm). This standard import way (used in SKRIPS) is referred to as Direct Import in Figure 4 (b), where the path of a variable X is shown. X refers to any component of the heat flux: latent, sensible, shortwave and longwave next fluxes. In the Direct import, $X_{PWRF}$ is imported as is into the MITgcm exf package. However, we find out that the flux components have already been scaled in PWRF to account for mixed grid cells, partially occupied by sea ice. This occurs mid-timestep (module_surface_driver.F in PWRF, Figure 4 (a), $X_{scaled}$) and the model output contains these scaled variables. This implies that when MITgcm receives the variable and accounts for the presence of sea ice in its own routines, the heat fluxes are scaled twice, thereby affecting the amount of heat gained or spent over the sea ice areas. To tackle this issue, we have developed a new technique (referred to as Separate Fluxes), whereby the separate heat flux components are captured in PWRF over the sea ice and over the ocean separately (Figure 4 (a), $X_{ice}$ and $X_{ocean}$). In the Separate fluxes approach two variables are now imported for each heat flux component: $X_{ocean}$ becomes the exf variable; and $X_{ice}$ is used instead of the recalculation of fluxes (solve4temp.F in MITgcm) to determine the amount of heat available for sea ice growth/melt (F_ia variable).

**2.5. Scaling by the sea ice concentration mask**

The development of P-SKRIPS includes the import of the individual heat and mass flux components from PWRF into MITgcm. The import of surface fluxes in the external forcing package (exf package) is prohibited when using the sea ice package in a standard version of MITgcm, because the fluxes need to be calculated according to the sea ice mask. We have set up the two models with the same grid and the same ice-ocean mask between PWRF and MITgcm, so the fluxes can be directly used in the sea ice packages, without any mask mismatch. We thus remove this error flag from seaice_model.F (in MITgcm).

This standard import way (used in SKRIPS) consists in directly importing the PWRF output variable ($X_{PWRF}$) to MITgcm. When $X_{PWRF}$ refers to any component of the heat flux: latent, sensible, shortwave and longwave net fluxes or evaporation, it is scaled to account for mixed grid cells, partially occupied by sea ice in the model cell in PWRF (Figure 4 (a)). This occurs mid-timestep (surface_driver.f90 in PWRF) and the PWRF model output contains these scaled variables. The Direct Import variable path in Figure 4 (b) illustrates how this scaled $X_{PWRF}$ is then imported as is into the MITgcm exf package. This implies that when MITgcm receives the variable and accounts for the presence of sea ice in its own routines, the variables are scaled twice, thereby affecting the amount of heat gained or spent over the sea ice areas. In the P-SKRIPS version, we exchange the four heat flux components between the models for physical consistency. To tackle the double-scaling issue, we have developed a new technique (referred to as Separate Fluxes), illustrated in Figure 4 (c), whereby the separate heat flux components are captured in PWRF over the sea ice and over the ocean separately ($X_{ice}$ and $X_{ocean}$). In this new approach two variables are now imported from PWRF to MITgcm, for each heat flux component: $X_{ocean}$ becomes the exf variable; and $X_{ice}$ is used instead of the recalculation of fluxes (solve4temp.F in MITgcm) to determine the amount of heat available for sea ice growth/melt (F_ia variable). This ensures that the fluxes are calculated and scaled only once between the two models.

**Line 217: What is the exf package? Should give a brief explanation before talking about it.**

Thank you, the exf package refers to the external forcing package in MITgcm and its description can be found in the documentation: "The external forcing package, in conjunction with the calendar package (cal), enables the handling of real-time (or "model-time") forcing fields of differing temporal forcing patterns. It comprises climatological restoring and relaxation. Bulk formulae are implemented to convert atmospheric fields to surface fluxes. An interpolation routine provides on-the-fly interpolation of forcing fields an arbitrary grid onto the model grid."

We have added the full name "external forcing package (exf package)" the first time we mention it in the manuscript.

The import of surface fluxes in the external forcing package (exf package) is prohibited when using the sea ice package in a standard version of MITgcm

**Line 268: "In that regard, the land ice meltwater runoff is captured in PWRF." I am not sure what this sentence means. Please clarify.**

Apologies, this will be rephrased to improve the clarity of the message:

In that regard, the land ice meltwater runoff is captured in PWRF.

For that reason, we capture the land ice meltwater runoff and send it to MITgcm.

**Line 269: Comma needed on "i.e.,"**

Thank you. This will be adapted.

(i.e. the amount from that timestep minus the amount from the previous timestep)

(i.e., the amount from that timestep minus the amount from the previous timestep)

**Line 279: "Drygalski Ice Tongue"**

Correct.

Thus, we are missing meltwater runoff from the Drygalsky Ice Tongue

Thus, we are missing meltwater runoff from the Drygalski Ice Tongue

**Figure 8e: This difference in precipitation, which is explained by the authors as being due to the snow being double counted, is very large. However, I find the explanation for this confusing. They say that in SKRIPS, the total precipitation is the sum of RAINNCV and SNOWNCV. If I understand correctly, the issue is then that SNOWNCV is a component of RAINNCV already, and so adding them double-counts SNOWNCV. If this is correct, the wording needs to be changed to make this more clear. I think my main confusion stems from line 314, where it reads:**

**"...and, therefore, is accounted for twice in the snowfall term."**

**Shouldn't this say something like:**

**"...and, therefore, is accounted for twice in the total precipitation."?**

**Since the issue is that the snowfall itself is counted twice, not that something is counted twice within the snowfall term?**

Yes, we agree that it is not clear and the wording will be adapted in the new version of the manuscript.

However, the latter is defined as a component of the non-convective precipitation, which encompasses all species (rain, graupel and snow and ice) and, therefore, is accounted for twice in the snowfall term

However, the latter is defined as a component of the non-convective precipitation, which encompasses all species (rain, graupel and snow and ice) and, therefore, is accounted for twice in the precipitation term

**Relatedly, since this difference is so big, is this an issue in existing models that the authors have identified? If so, I would suggest making more of this since it surely has a large impact to have double the precipitation that it should.**

Thanks, we will add more details in the text.

However, the latter is defined as a component of the non-convective precipitation, which encompasses all species (rain, graupel and snow and ice) and, therefore, is accounted for twice in the snowfall term. In the P-SKRIPS, the components are added individually and the time step non convective precipitation term is ignored.

However, the latter is defined as a component of the non-convective precipitation, which encompasses all species (rain, graupel and snow and ice) and, therefore, is accounted for twice in the precipitation term. It is likely that over the course of the winter season, additional accumulation of snow over the sea ice affect the heat transfer through that layer, and in spring it will lead to increased freshwater flux into the surface of the ocean. In the P-SKRIPS, the components are added individually and the time step non convective precipitation term is ignored.

**Line 337: "This is an improvement…". An improvement over what? Be more clear about what you are comparing to here.**

Thank you. We will refine the text to:

One-month test experiments for the January and August of 2016 have been carried out and show that the P-SKRIPS approach homogenises the ocean and atmosphere calculations. This is an improvement and is needed for balanced long term simulations, especially in coastal polynyas.

One-month test experiments for the January and August of 2016 have been carried out and show that the P-SKRIPS approach homogenises the ocean and atmosphere calculations. This is an improvement to pre-existing ocean-atmosphere coupling in regional climate models, and is needed for balanced long term simulations, especially in coastal polynyas.

**Line 340-341: "Snow cover changes happen in PWRF. Its multi-layered snowpack and complex representation of snow on sea ice makes PWRF the better model to represent…".**

**This is new, right? If I understand correctly in the previous version of the model, snow cover changes were done in both MITgcm and PWRF and so were inconsistent. Make it clear that this is new, and a change from the old way of doing it. You say it makes PWRF the "better model". As with my comment above, be clear about what it is better than specifically.**

Yes, the previous version of the model did not exchange the snow cover, and both models made it evolve independently, and so

were inconsistent. We will clarify both this and "better model" in the new version of the text:

The formation and advection of sea ice stay in MITgcm. Snow cover changes happen in PWRF. Its multi-layered snowpack and complex representation of snow on sea ice makes PWRF the better model to represent the snow/ice-atmosphere interactions. P-SKRIPS also splits the processes related to sea ice between the two models. The formation and advection of sea ice stay in MITgcm. The snow cover changes were happening in both PWRF and MITgcm in the previous version of the model, leading to inconsistent snow on sea ice. In P-SRIPS, the snow cover changes only happen in PWRF, and are then sent to MITgcm. Its multi-layered snowpack and complex representation of snow on sea ice makes PWRF more suitable to represent the snow/ice-atmosphere interactions.

**Line 354: parameterization -> parameterizations**

This will be corrected.

In summer, large differences in shortwave fluxes are directly linked to the calculation of the albedo, which varies between the model components as they have different parameterization of dry/wet snow.

In summer, large differences in shortwave fluxes are directly linked to the calculation of the albedo, which varies between the model components as they have different parameterizations of dry/wet snow.

**Line 378: "While the atmosphere's response to ..."**

This will be corrected.

While the atmosphere's response the extra energy is the formation of localized weather event

While the atmosphere's response to the extra energy is the formation of localized weather event

**Line 380: "The most accurate representation..."**

**I do not agree with the use of the word accurate here, since you are not evaluating the model against observations, so you have no basis on which to say it is more accurate. I would suggest saying "most physically consistent" or similar.**

We thank you for this suggestion, that will be implemented in the new version of the manuscript.

The presented coupled model setup constitutes, to our knowledge, the most accurate representation of ocean/atmosphere/sea ice interactions for polar climates and is thus recommended for climate modelling in any Arctic and Antarctic region.

The presented coupled model setup constitutes, to our knowledge, the most physically consistent representation of ocean/atmosphere/sea ice interactions for polar climates and is thus recommended for climate modelling in any Arctic and Antarctic region.

**As noted in the existing Discussion comment, the authors have not yet followed the journal guidelines on reproducibility, and need to do so before publication.**

We are aware of this problem. We have provided all of the files that are under open access licences to the repositories: https://doi.org/10.5281/zenodo.7739063 and https://doi.org/10.5281/zenodo.7739059. Unfortunately, the PWRF model ( https://doi.org/10.1029/2012J018139) files are not publicly available for publishing and the decision is beyond our control. However,

the files can be obtained on the PWRF website upon request https://polarmet.osu.edu/PWRF/ . We are also able to provide files to the reviewers for the purpose of reviewing this manuscript through the editor if needed. The data availability statement has been updated accordingly.

The developments and files required to set up and run the model presented in this paper are available at https://github.com/alena-malyarenko/P-SKRIPS and on Zenodo https://doi.org/10.5281/zenodo.7297744. The coupled model builds on the Scripps-Kaust model described in Cerovecki et al. (2022) and Sun et al. (2019). The base code can be find at https://github.com/iurnus/scripps_kaust_model/. ERA5 Reanalysis (0.25 Degree Latitude-Longitude Grid), generated by European Centre for Medium-Range Weather Forecasts are available at https://cds.climate.copernicus.eu/cdsapp#!/dataset/reanalysis-era5-pressure-levels and https://cds.climate.copernicus.eu/cdsapp#!/dataset/reanalysis-era5-single-levels. BSOSE data is availbe at http://sose.ucsd.edu/. The Bedmap 2 data can be downloaded from https://www.bas.ac.uk/project/bedmap-2/.

Our model code for the Ross Sea case is based on PolarWRF and MITgcm. The P-SKRIPS model code is an updated version of SKRIPS (https://doi.org/10.5281/zenodo.7336070) and can be found in the two directories due to size limits (https://doi.org/10.5281/zenodo.7739063 and https://doi.org/10.5281/zenodo.7739059). The detailed instructions for converting WRF to PWRF ( https://doi.org/10.1029/2012J018139) can be found within PSKRIPS-main folder. The short description of steps for PWRF is as follows: find PSKRIPS/Models/WRF_4.1.3 within the repository; obtain PWRF-4.1.3 modifications by email and merge the code locally; add P-SKRIPS modifications to PWRF; compile.

ERA5 Reanalysis (0.25 Degree Latitude-Longitude Grid), generated by European Centre for Medium-Range Weather Forecasts are available at https://cds.climate.copernicus.eu/cdsap

p#!/dataset/reanalysis-era5-pressure-levels and https://cds.climate.copernicus.eu/cdsapp#!/dataset/reanalysis-era5-single-levels. BSOSE data is availbe at http://sose.ucsd.edu/. The Bedmap 2 data can be downloaded from https://www.bas.ac.uk/project/bedmap-2/.

References:

Smith, R. S., Mathiot, P., Siahaan, A., Lee, V., Cornford, S. L., Gregory, J. M., et al. (2021). Coupling the U.K. Earth System model to dynamic models of the Greenland and Antarctic ice sheets. Journal of Advances in Modeling Earth Systems, 13, e2021MS002520. https://doi.org/10.1029/2021MS002520

Bozkurt, D., Bromwich, D.H., Carrasco, J. et al. Temperature and precipitation projections for the Antarctic Peninsula over the next two decades: contrasting global and regional climate model simulations. Clim Dyn 56, 3853–3874 (2021). https://doi.org/10.1007/s00382-021-05667-2

Wang, W., Bruyère, C., Duda, M., Dudhia, J., Gill, D., Kavulich, M., Werner, K., Chen, M., Lin, H.-C., Michalakes, J., Rizvi, S., Zhang, X., Berner, J., Munoz-Esparza, D., Reen, B., Ha, S., and Fossell, K.: WRF users' guide, https://www2.mmm.ucar.edu/wrf/users/docs/user_guide_v4/v4.1/contents.html, 2019.

Azaneu, M., Kerr, R., and Mata, M. M.: Assessment of the representation of Antarctic Bottom Water properties in the ECCO2 reanalysis, Ocean Sci, 10, 923–946, https://doi.org/10.5194/os-10-923-2014, 2014.

---

## Author Comment (AC4)

Author response to the Referee Comments by Anonymous Reviewer 2 on the manuscript:

**Conservation of heat and mass in P-SKRIPS version 1: the coupled atmosphere-ice-ocean model of The Ross Sea**

Alena Malyarenko, Alexandra Gossart, Rui Sun, and Mario Krapp

submitted to Geoscientific Model Developments (https://doi.org/10.5194/egusphere-2022-1135)

We thank the Reviewer for all the time and effort put into the review of our manuscript and are pleased with their positive and constructive comments. Please find the response to each of the comments below. The reviewer's comments are displayed in **bold text**, replies are shown in normal text, text from the original manuscript is shown in blue, and proposed changes to the

5 manuscript are shown in red.
* * *
**This is my first review of the manuscript entitled "Conservation of heat and mass in P-SKRIPS version 1: the coupled atmosphere-ice-ocean model of the Ross Sea". The authors first introduce the importance of regional climate modelling**

10 **in Antarctica. They introduce various existing setups and explain the challenges related to energy conservation in these setups. Then, they present their setup, which they call P-SKRIPS, which is based on the SKRIPS model, with a focus on the Ross and a particular attention to the conservation of energy. They describe how their implementation improves the consistency and conservation of heat and mass fluxes between the climate (PWARF) and the ocean (MITgcm) components of the coupled system. They compare the SKRIPS and the P-SKRIPS over two months (one in winter, one in**

15 **summer) and describe the impact of the improved flux conservation on the evolution of these fluxes.**

**The manuscript is generally well-written and pleasant to follow. The implementation and the setup are well-described and well-motivated. It is a significant contribution to the development of polar regional climate modelling and I would therefore recommend this manuscript for publication in GMD after my comments, generally minor, have been ad-**

20 **dressed.**

**1 Specific comments:**

**The abstract is short and to the point (congratulations!). However, I find that the expression "shows the advantages" is overstating the results of the manuscripts. This is at least a bit misleading, as I was expecting a more in-depth discussion of the scientific implication of heat conservation in the setup. I would suggest using "impacts" or "implications".**

25 We thank you for this remark and appreciate your appraisal of our abstaract. We will change the text accordingly.

P-SKRIPS v.1 shows the advantages of conserving heat flux over the Terra Nova Bay and Ross Sea polynyas in August 2016,

eliminating the mismatch between total flux calculation in PWRF and MITgcm up to 922 W m$^{-2}$.

P-SKRIPS v.1 shows the implications of conserving heat flux over the Terra Nova Bay and Ross Sea polynyas in August 2016, eliminating the mismatch between total flux calculation in PWRF and MITgcm up to 922 W m$^{-2}$.

**L57: It looks like there is an outdated Table reference in the manuscript file.**

We thank you for highlighting this. This is an error as no table is planned to be inserted here, we will remove the reference to the table.

**L69: "This is critical to avoid..." The introduction presents a good motivation for the study and is generally very well referenced, except maybe for this sentence (that is actually quite key to the focus of the study). I would recommend developing a bit on the importance of heat conservation. For instance, it is better when everything is conserved, but depending on the question that one tries to answer, this is not always critical. Can the authors show examples (ideally with references) where it is absolutely critical?**

We agree that 'critical' might have been an overstatement. We propose to add some references and to soften the statement to:

This is critical to avoid model drift and inconsistencies between the two model components over long term simulations.

This is necessary when modelling long-term air-ocean fluxes and interactions, in order to avoid model drift and inconsistencies between the two model components. For example, the amount of heat transferred form the atmosphere into the ocean or sea ice can have an impact on upwelling (Morrison et al. 2015; Skinner et al. 2020) and sea ice extent (Cerovecki 2022), as well as impacting the atmospheric boundary layer processes (e.g. Alam and Curry 1995).

**L116: "as is ordinary the case". This sentence is a bit confusing, ordinary for who? Do the authors mean a standard preprocessing described in WRF manual for instance? Do the authors have a reference for this?**

We agree that this is not clear. The WRF manual indicates a series of standard steps to manipulate the original input files so that they can be ingested by the WRF program. WRF users are probably familiar with this, but we will include more details in the manuscript.

The preprocessing of the input files was conducted as is ordinarily the case,

The preprocessing of the WRF input files followed the standard procedure (ungrib.exe - metgrid.exe - real.exe, Wang et al. (2019)),

**Figure 3: In the caption, I think there is a mistake with the references to the a,b,c,d panels (SKRIPS and P-SKRIPS are inverted).**

Yes, thank you for pointing this out. This will be corrected.

**L168: "The coupling ... ice sheet." I am sorry but I do not understand the end of this sentence.**

The two models simulate different parts of the ocean-sea ice, snow and atmosphere components: the oceanic realm is in the

ocean model, and the atmosphere is dealt with by the atmosphere model. Sea ice is composed of the ice floating on the ocean, and the snow layer on top of this block. The sea ice block changes and movement are dealt with in the ocean model, while the atmosphere model is responsible for the changes in snow cover (accumulation, compaction etc...) so the exchange interface

65  between the two models is between the surface of the sea ice block and the snowpack that lays on top of it. Over the ice sheet, there is no ocean but there is a snowpack/firn, overlying pure ice. Typically an ice sheet model would take care of the ice dynamics (not present in this configuration) and the atmosphere model deals with snowpack changes (meltwater runoff for instance). In P-SKRIPS, this runoff is then exiting into the ocean. We agree that the ice sheet part (last part of the sentence) is confusing and probably not needed. We propose to rephrase as follows:

70  The coupling interface between models is defined as air-ocean over open ocean, between the sea ice and the snow on top of it, and below the snowpack on top of the ice sheet

The coupling interface between the atmosphere and ocean models is defined as (1) air-ocean over open ocean, (2) between the sea ice (changes and advection of sea ice simulated by the ocean model) and the snow layer on top of it (accumulation and compaction are dealt with by the atmosphere model).

75

**L169: This paragraph seems to repeat some statements made in the previous paragraphs, statements that are re-peated L208. This repetition, instead of clarifying things, makes them a bit more confusing in my opinion. Was this paragraph intended to be a short summary of the subsection? If yes, I would recommend either shortening it to maximum of 2 short sentences or removing it. (Same for L208–>214.)**

80  We thank the reviewer for this remark. This is indeed repetitions that can be removed. We propose to keep the first paragraph, and remove the subsequent ones.

**L217: I cannot find the introduction of the "exf" package before in the text. I would recommend giving a bit of context to the reader of what this package is.**

85  Thank you, the exf package refers to the external forcing package in MITgcm and its decription can be found in the documentation: "The external forcing package, in conjunction with the calendar package (cal), enables the handling of real-time (or "model-time") forcing fields of differing temporal forcing patterns. It comprises climatological restoring and relaxation. Bulk formulae are implemented to convert atmospheric fields to surface fluxes. An interpolation routine provides on-the-fly interpolation of forcing fields an arbitrary grid onto the model grid." We have added the full name "external forcing package

90  (exf package)" the first time we mention it in the manuscript.

The import of surface fluxes in the external forcing package (exf package) is prohibited when using the sea ice package in a standard version of MITgcm

**L218: "In the coupled setup": The manuscript mostly focuses on coupled setups. Which one is referred to here?**

95  We refer to our way of defining the model domains, that was set up to not to have to regrid or interpolate and have corresponding grids and the same ocean-land mask in the two models. This is the case, regardless of whether we use SKRIPS or

P-SKRIPS.

In the coupled set up, the sea ice mask is coordinated between PWRF and MITgcm, so the fluxes can be directly used in the sea ice packages, without any mask mismatch.

100  We have set up the two models with the same grid and the same ice-ocean mask between PWRF and MITgcm, so the fluxes can be directly used in the sea ice packages, without any mask mismatch.

**L218: Not sure of what the authors mean with "coordinated"**

As mentioned in the previous comment, we have 'coordinated' our efforts in setting up the models so that both have exactly

105  the same grid, and the same ice-ocean mask. We will rephrase in the new version of the manuscript:

In the coupled set up, the sea ice mask is coordinated between PWRF and MITgcm, so the fluxes can be directly used in the sea ice packages, without any mask mismatch.

We have set up the two models with the same grid and the same ice-ocean mask between PWRF and MITgcm, so the fluxes can be directly used in the sea ice packages, without any mask mismatch.

110

**Figure 6: The colorbar saturates a lot for the differences, so it is hard to believe there are only "subtle differences in the order of $10^{-3}W.m^{-2}$". There are weird also patterns (vertical lines north of the ice shelf) that are not well explained. Do the authors have an explanation for them?**

The stripes are present over areas of low sea ice concentration (below 0.1 sea ice cover) and only in the first 10 days of the

115  simulation. We attribute these to the spinup adjustment of the two models to the mismatch in the input data for sea ice cover and sea surface temperature: coming from ERA-5 for PWRF and from BSOSE for MITgcm. We have updated the figure caption to include the full range of the differences and include the figure with updated range for information below. We decided to keep the original figure in the text because we focus on the heterogeneity caused by the sea ice mask difference, which is close to $10^{-3}W.m^{-2}$.

120  (a) Latent Heat (LH) flux: total heat flux in PWRF using the capture of the Separate fluxes, (b) latent heat in MITgcm, (c) differences between (a) and (b) and (d) ice concentration mask difference between the two timesteps. See Section 3.1.2 for more details.

(a) Latent Heat (LH) flux: total heat flux in PWRF using the capture of the Separate fluxes, (b) latent heat in MITgcm, (c) differences between (a) and (b); (d) ice concentration mask difference between the two timesteps. The full range of differences

 for (c) is -0.08 to 0.02 $W.m^{-2}$. See Section 2.5 for more details.

[Figure]

[Figure]

[h]

**L241: 104-->$10^4$ I suppose?**

Correct. We will change this.

the scaling multiplication by $X_{ICE}$, which is 104

130    the scaling multiplication by $X_{ICE}$, which is $\sim 10^4$

**Table S1 is very unclear. Please add context to its caption. Why is there a "Total" row, and why is it empty?**

We agree that the caption is not clear. The table is meant to indicate the definition of each flux component, in each model (positive upward or downward). The TOTAL column is indeed not needed. We will update the Table accordingly.

135

**"At midday... (Figure S4)". I do not understand the link between this comment on the flux evolution and the rest of the paragraph that discusses discrepancies in MITgcm fluxes between the two setups. I do not see larger discrepancies associated with these peaks.**

We agree and propose to remove the sentence.

140

**Figure S4 could belong to the main manuscript in my opinion. The same for Table 2. Just a suggestion.**

We thank you for this suggestion, but we prefer to have them in the supplements.

**Table 1.** Import of heat and mass fluxes versus calculations in the two experiments

| | | MITgcm exf | MITgcm seaice | PWRF |
|---|---|---|---|---|
| HEAT | latent heat | ↓ | ↑ | ↑ |
| | sensible heat | ↓ | ↑ | ↑ |
| | short wave net | ↑ | ↓ | ↓ |
| | long wave net | ↑ | ↑ | ↓ |
| | TOTAL | ↑ | ↑ | - |
| MASS | evaporation | ↑ | - | ↑ |
| | precipitation | ↓ | - | ↓ |
| | sea ice runoff | ↓ | - | ↓ |
| | land runoff | ↓ | - | ↓ |
| | TOTAL | | | |

**Table 2.** Sign definition of each flux, in each model component. ↓ indicates defined positive downward, ↑ indicates defined positive upward.

| | | MITgcm exf | MITgcm seaice | PWRF |
|---|---|---|---|---|
| HEAT | latent heat | ↓ | ↑ | ↑ |
| | sensible heat | ↓ | ↑ | ↑ |
| | short wave net | ↑ | ↓ | ↓ |
| | long wave net | ↑ | ↑ | ↓ |
| MASS | evaporation | ↑ | - | ↑ |
| | precipitation | ↓ | - | ↓ |
| | sea ice runoff | ↓ | - | ↓ |
| | land runoff | ↓ | - | ↓ |

**I find Figure 8 and S4 not very readable. I get why the authors want to show both the evolution of the fluxes and the differences between MITgcm and PWRF but doing both on the same figures means these differences are hard to see. I would really recommend plotting the differences on a different figure. At the very least, the authors could play with the width of the different lines, or add markers, to make it easier to distinguish what is what.**

We thank you for this suggestion, we will add two new figures showing the differences and refer to them in the text, as appropriate. In addition, we will add two supplementary tables indicating the mean value for each of these timeseries and the differences.

[revised manuscript text omitted]

**L310–>318: This paragraph seems quite important (at least to me), but I find it quite confusing. I would strongly recommend rephrasing it. The differences between the two setups are particularly significant for this term, so it is worth clarifying what the implications are.**

We thank you for this comment, in line with Reviewer 1 comment. We propose to clarify as follows:

**Table 3.** Statistics presenting the mean value for the different variables in Figures 8 and S4 in January for both the SKRIPS and the P-SKRIPS simulations, as well as the mean values for the differences between the PWRF and the MITgcm variables for each of these simulations. The variables are integrated over the whole simulation and through the entire domain.

| simulation JAN | LH [W] | SH [W] | LWNET [W] | SWNET [W] | Prec. $[m^3 s^{-1}]$ | Evap. $[m^3 s^{-1}]$ | Runoff $m^3 s^{-1}$ |
|---|---|---|---|---|---|---|---|
| SKRIPS | $2.69e^{13}$ | $1.68e^{13}$ | $8.87e^{13}$ | $-3.93e^{14}$ | $7.74e^4$ | $1.03e^4$ | PWRF $1.09e^4$ |
| P-SKRIPS | $2.44e^{13}$ | $1.20e^{13}$ | $9.19e^{13}$ | $-3.92e^{14}$ | $3.69e^4$ | $9.38e^3$ | MITgcm $1.13e^4$ |
| SKRIPS difference | $-2.74e^{12}$ | $-2.89e^{12}$ | $1.71e^{13}$ | $1.e^{13}$ | $-0.0036$ | $5.33e^{-5}$ | diff $7.18e^{-5}$ |
| P-SKRIPS difference | $7.87e^7$ | $2.38e^8$ | $-2.64^{11}$ | $1.44e^{11}$ | $-0.0019$ | $-3.40e^{-5}$ | |

**Table 4.** Statistics presenting the mean value for the different variables in Figures 8 and S4 in August for both the SKRIPS and the P-SKRIPS simulations, as well as the mean values for the differences between the PWRF and the MITgcm variables for each of these simulations. The variables are integrated over the whole simulation and through the entire domain.

| simulation AUG | LH [W] | SH [W] | LWNET [W] | SWNET [W] | Prec. $[m^3 s^{-1}]$ | Evap. $[m^3 s^{-1}]$ | Runoff $m^3 s^{-1}$ |
|---|---|---|---|---|---|---|---|
| SKRIPS | $2.06e^{13}$ | $-5.83e^{12}$ | $1.24e^{14}$ | $-9.19e^{12}$ | $7.02e^4$ | $8.46e^3$ | PWRF 0.99 |
| P-SKRIPS | $7.42e^{12}$ | $-1.69e^{13}$ | $1.14e^{14}$ | $-8.21e^{12}$ | $3.27e^4$ | $3.34e^3$ | MITgcm 0.99 |
| SKRIPS difference | $-1.77e^{11}$ | $-8.24e^{10}$ | $1.64e^{13}$ | $-1.14e^{12}$ | $-0.0029$ | $-1.1e^{-5}$ | diff $-4.15e^{-9}$ |
| P-SKRIPS difference | $5.85e^7$ | $1.46e^8$ | $-1.29e^{12}$ | $6.24e^8$ | $-9.71e^{-4}$ | $6.84e^{-6}$ | |

However, the latter is defined as a component of the non-convective precipitation, which encompasses all species (rain, graupel and snow and ice) and, therefore, is accounted for twice in the snowfall term. In the P-SKRIPS, the components are added individually and the time step non convective precipitation term is ignored.

However, the latter is defined as a component of the non-convective precipitation, which encompasses all species (rain, graupel and snow and ice) and, therefore, is accounted for twice in the precipitation term. It is likely that over the course of the winter season, additional accumulation of snow over the sea ice affect the heat transfer through that layer, and in spring it will lead to increased freshwater flux into the surface of the ocean. In the P-SKRIPS, the components are added individually and the time step non convective precipitation term is ignored.

**L377: A bit in line with my first comment, it is likely that these differences in heat fluxes will affect these processes. However, it has not been proven in the manuscript and I would therefore recommend rephrasing a bit. For instance, use "likely" instead of "directly".**

We agree that this is a limitation in our technical paper. We will explore more of the fluxes in a subsequent "scientific" paper. We will therefore rephrase:

The non-conservation of up to 922 W m$^{-2}$ is directly affecting the heat content of the atmosphere and deep convection of the ocean.

The non-conservation of up to 922 W m$^{-2}$ is likely affecting the heat content of the atmosphere and deep convection of the

ocean.

**L380: This sentence sounds like an overstatement. What are the metrics used by the authors to make such a claim? Depending on the question that is asked, another model may be more suitable and have a much better representation**

245 **of the processes of interest. It is a nice paper overall, do not upset readers that would only read the conclusion!**

We thank the reviewer for this fair comment. We will change our conclusive sentence according to comments from both reviewers.

The presented coupled model setup constitutes, to our knowledge, the most accurate representation of ocean/atmosphere/sea ice interactions for polar climates and is thus recommended for climate modelling in any Arctic and Antarctic region.

250 The presented coupled model setup constitutes, to our knowledge, the most physically consistent representation of ocean/atmosphere/sea ice interactions for polar climates and is thus recommended for climate modelling in any Arctic and Antarctic region.

**I wish the authors good luck with the revisions!**

Thank you :)

255

**comment from the authors:** Note that as noted in the existing Discussion comment, part of the code is not in open access. We have provided all of the files that are under open access licences to the repositories: https://doi.org/10.5281/zenodo.7739063 and https://doi.org/10.5281/zenodo.7739059. Unfortunately, the PWRF model ( https://doi.org/10.1029/2012J018139) files are not publicly available for publishing and the decision is beyond our control. However, the files can be obtained on the PWRF

260 website upon request https://polarmet.osu.edu/PWRF/ . We are also able to provide files to the reviewers for the purpose of reviewing this manuscript through the editor if needed. The data availability statement has been updated accordingly.

The developments and files required to set up and run the model presented in this paper are available at https://github.com/alena-malyarenko/P-SKRIPS and on Zenodo https://doi.org/10.5281/zenodo.7297744. The coupled model builds on the Scripps-Kaust model described in Cerovecki et al. (2022) and Sun et al. (2019). The base code can be find at https://github.com/iurnus/

265 scripps_kaust_model/. ERA5 Reanalysis (0.25 Degree Latitude-Longitude Grid), generated by European Centre for Medium-Range Weather Forecasts are available at https://cds.climate.copernicus.eu/cdsapp#!/dataset/reanalysis-era5-pressure-levels and https://cds.climate.copernicus.eu/cdsapp#!/dataset/reanalysis-era5-single-levels. BSOSE data is availbe at http://sose.ucsd .edu/. The Bedmap2 data can be downloaded from https://www.bas.ac.uk/project/bedmap-2/.

Our model code for the Ross Sea case is based on PolarWRF and MITgcm. The P-SKRIPS model code is an updated version

270 of SKRIPS (https://doi.org/10.5281/zenodo.7336070) and can be found in the two directories due to size limits (https://doi.org/ 10.5281/zenodo.7739063 and https://doi.org/10.5281/zenodo.7739059). The detailed instructions for converting WRF to PWRF ( https://doi.org/10.1029/2012J018139) can be found within PSKRIPS-main folder. The short description of steps for PWRF is as follows: find PSKRIPS/Models/WRF_4.1.3 within the repository; obtain PWRF-4.1.3 modifications by email and merge the code locally; add P-SKRIPS modifications to PWRF; compile.

275 ERA5 Reanalysis (0.25 Degree Latitude-Longitude Grid), generated by European Centre for Medium-Range Weather Forecasts

are available at https://cds.climate.copernicus.eu/cdsapp#!/dataset/reanalysis-era5-pressure-levels and https://cds.climate.cope-rnicus.eu/cdsapp#!/dataset/reanalysis-era5-single-levels. BSOSE data is availabe at http://sose.ucsd.edu/. The Bedmap 2 data can be downloaded from https://www.bas.ac.uk/project/bedmap-2/.

---

## Referee Report (RR1)

**Second Review of "Conservation of heat and mass in P-SKRIPS version 1: the coupled atmosphere-ice-ocean model of The Ross Sea" by Alena Malyarenko, Alexandra Gossart, Rui Sun and Mario Krapp.**

The authors have submitted a much improved manuscript, and I believe it to be suitable for publication after addressing the very minor comments below:

**Minor Comments:**

Line 29: "due to the remoteness **of** and harsh conditions prevailing in the Antarctic."

Line 33: "Warmer **global**-mean surface temperartures lead to..."

Line 37-53: The acronyms "ESM" and "GCM" are used interchangeably here. In general for the context here they are the same thing, but I would suggest sticking with one to avoid confusion.

Also, Hines and Bromwich (2008) is an odd choice of citation here (Line 39). This paragraph is talking about how global models are not optimized for polar regions, but the citation is for a paper presenting results from a polar-optimized regional model (PWRF). Would suggest finding a better reference for this statement or removing.

Line 61: "However, fully-coupled ocean-atmosphere-sea ice (and ice sheet) **models** are rare for any of the..."

Line 83: Here and in other places there are parentheses around the year for references already in parentheses. I suspect this will be caught in copy-editing though.

Line 102: "... are extracted from **the** BEDMAP-2 product (Fretwell et al., 2013)."

Line 118: dissociates -> distinguishes

Line 121: "...and has a refined snow water equivalent reproduction (closer to reality)." I appreciate that the authors have revised this sentence, however my question was really about what IS reality here? What observational product is the model being compared to to make this assessment?

Line 124: 61 model levels -> 61 vertical levels

Line 149: "The experiments were set with the aim of testing the model skill..."
Similar to my first round of comments, you are not really testing model skill. That is, you are not evaluating the model's ability to reproduce observations or another model. Would suggest rephrasing to "...with the aim of comparing various model configurations for different sea ice cover conditions (winter vs summer)."

Line 151: "cast study" -> "case study"

Line 197: "This standard import way" -> "This standard import method"

Line 334-335: "...the components are added individually and the time step non conveective precipitation term is ignored."

I think the words "time step" are not needed here? So I think this should be "...the components are added individually and the non-convective precipitation term is ignored."

Line 362: P-SRIPS -> P-SKRIPS

Line 398-399: Would suggest changing to "The energy imbalance of up to 922 W m$^{-2}$ due to non-conservation of energy in the previous Skripps-KAUST version of the model likely affects the heat content ..." or similar.

I would also like to note that I am pleased to see that a solution has hopefully been come to for the issue of code availability.

---

## Author Response (AR2)

Author response to the Referee Comments by Anonymous Reviewer 1 on the manuscript:

**Conservation of heat and mass in P-SKRIPS version 1: the coupled atmosphere-ice-ocean model of The Ross Sea**

Alena Malyarenko, Alexandra Gossart, Rui Sun, and Mario Krapp

submitted to Geoscientific Model Developments (https://doi.org/10.5194/egusphere-2022-1135)

We thank the Reviewer for spotting all the typos and inconsistencies of our manuscript and are pleased with their positive and constructive comments. Please find the response to each of the comments below. The reviewer's comments are displayed in **bold text**, replies are shown in normal text, text from the original manuscript is shown in blue, and proposed changes to the manuscript are shown in red.
* * *
**The authors have submitted a much improved manuscript, and I believe it to be suitable for publication after addressing the very minor comments below:**

**Line 29: "due to the remoteness of and harsh conditions prevailing in the Antarctic."**

Thank you, we will add 'of'.

due to the remoteness and harsh conditions prevailing in the Antarctic.

due to the remoteness of and harsh conditions prevailing in the Antarctic.

**Line 33: "Warmer global-mean surface temperartures lead to..."**

Thank you, we will add 'global'.

Warmer mean surface temperatures lead to

Warmer global mean surface temperatures lead to

**Line 37-53: The acronyms "ESM" and "GCM" are used interchangeably here. In general for the context here they are the same thing, but I would suggest sticking with one to avoid confusion.**

Yes, it is indeed quite confusing for the reader. We will change all the acronyms to 'ESMs' and will add a line specifying that by ESMs we also mean GCMs.

Earth System Models (ESMs) and Global Circulation Models (GCMs) present an alternative to explore the past, present and future state of Antarctica and the Southern Ocean. Studies using GCMs or ESMs have highlighted the importance of atmosphere-ocean-sea ice interactions in the representation of polar systems, especially for estimating the evolution of Antarctica in a future, warmer setting (e.g. Goosse et al., 2018). Warmer mean surface temperatures lead to increased basal melting

of ice shelves (Naughten et al., 2021). In addition, warmer ocean water masses in cavities can be induced by increased surface stress due to thinning of the sea ice (Hellmer et al., 2012) and increased incoming radiation causing surface melting. The latter can lead to ice shelf fragilisation and potential collapse (DeConto and Pollard, 2016).

ESMs that are part of the Coupled Model Intercomparison Project (CMIP) experiments, generally have coupled global ocean, atmosphere, land and sea ice models (Meehl et al., 1997). However, the global atmosphere and ocean models that make up ESMs are not optimized for polar areas (e.g. Azaneu et al., 2014) and polar versions of these models are developed to represent processes specific to these regions. In addition, the spatial resolution of ESMs is rather coarse, which prevents them from representing local or regional-scale processes. For example, Smith et al. (2021) raises the fact that accumulation and melt at the ice-ocean-atmosphere interface have refined spatial patterns that can not be represented in GCMs. And this leads to static ice boundaries and heavy parametrization of these processes, limiting the inclusion of refined ice sheet or ice shelf cavity models into GCMs. Therefore, ice-ocean and ice-atmosphere interactions are usually not accurately representend into GCMs. In addition, the parametrization of processes occurring at higher resolution in GCMs physics limits them in the representation of local scale and regional features (e.g., the orography and associated local processes of the Antarctic Peninsula, Bozkurt et al., 2021), indicating that the global physics of GCMs are not optimised for polar areas (Agosta et al., 2015; Bozkurt et al., 2021), leading to various performances in the Arctic and Antarctic.

Earth System Models and Global Circulation Models (ESMs and GCMs respectively, for simplicity we will refer to both as ESMs hereafter) present an alternative to explore the past, present and future state of Antarctica and the Southern Ocean. Studies using ESMs have highlighted the importance of atmosphere-ocean-sea ice interactions in the representation of polar systems, especially for estimating the evolution of Antarctica in a future, warmer setting (e.g. Goosse et al., 2018). Warmer global mean surface temperatures lead to increased basal melting of ice shelves (Naughten et al., 2021). In addition, warmer ocean water masses in cavities can be induced by increased surface stress due to thinning of the sea ice (Hellmer et al., 2012) and increased incoming radiation causing surface melting. The latter can lead to ice shelf fragilisation and potential collapse (DeConto and Pollard, 2016).

ESMs that are part of the Coupled Model Intercomparison Project (CMIP) experiments, generally have coupled global ocean, atmosphere, land and sea ice models (Meehl et al., 1997). However, the global atmosphere and ocean models that make up ESMs are not optimized for polar areas (e.g. Azaneu et al., 2014) and polar versions of these models are developed to represent processes specific to these regions. In addition, the spatial resolution of ESMs is rather coarse, which prevents them from representing local or regional-scale processes. For example, Smith et al. (2021) raises the fact that accumulation and melt at the ice-ocean-atmosphere interface have refined spatial patterns that can not be represented in ESMs. And this leads to static ice boundaries and heavy parametrization of these processes, limiting the inclusion of refined ice sheet or ice shelf cavity models into ESMs. Therefore, ice-ocean and ice-atmosphere interactions are usually not accurately representend into ESMs. In addition, the parametrization of processes occurring at higher resolution in ESMs physics limits them in the representation of local scale and regional features (e.g., the orography and associated local processes of the Antarctic Peninsula, Bozkurt et al., 2021), indicating that the global physics of ESMs are not optimised for polar areas (Agosta et al., 2015; Bozkurt et al.,

2021), leading to various performances in the Arctic and Antarctic.

**Also, Hines and Bromwich (2008) is an odd choice of citation here (Line 39). This paragraph is talking about how global models are not optimized for polar regions, but the citation is for a paper presenting results from a polar-optimized regional model (PWRF). Would suggest finding a better reference for this statement or removing**

We thank you for this remark and will remove the citation.

are not optimized for polar areas (e.g. Hines et al., 2008; Azaneu et al., 2014) and polar versions of these models are developed to represent processes specific to these regions.

are not optimized for polar areas (e.g. Azaneu et al., 2014) and polar versions of these models are developed to represent processes specific to these regions.

**Line 61: "However, fully-coupled ocean-atmosphere-sea ice (and ice sheet) models are rare for any of the..."**

Thank you, we will add 'models'.

However, fully coupled ocean-atmosphere-sea ice (and ice sheets) are rare for any of the regional models

However, fully coupled ocean-atmosphere-sea ice (and ice sheets) models are rare for any of the regional models

**Line 83: Here and in other places there are parentheses around the year for references already in parentheses. I suspect this will be caught in copy-editing though.**

We thank you for spotting this and will edit all the citations accordingly.

**Line 102: "... are extracted from the BEDMAP-2 product (Fretwell et al., 2013)."**

Thank you, we will add 'the'.

The model bathymetry, ice shelf draft, and grounding line are extracted from BEDMAP-2 product

The model bathymetry, ice shelf draft, and grounding line are extracted from the BEDMAP-2 product

**Line 118: dissociates -> distinguishes**

Thank you, we will change 'dissociates' by 'distinguishes'.

and dissociates between bare ground, land ice (glaciers) and sea ice.

and distinguishes between bare ground, land ice (glaciers) and sea ice.

**Line 121: "...and has a refined snow water equivalent reproduction (closer to reality)." I appreciate that the authors have revised this sentence, however my question was really about what IS reality here? What observational product is the model being compared to to make this assessment?**

The new version of the Noah-MP model has been tested against direct measurements of snow depth, SWE, snow surface albedo and snow skin temperature. These were taken at the Sleeper River Watershed (Vermont, USA) and Col de Porte in the French

Alps. This is described in Niu et al., 2011 and Figure 5.b) shows an improvement of SWE between the older version of Noah, and the new one for the Sleeper River Watershed. Figures 6 and 7 illustrate the improvement for SWE and snow depth at Col de Porte.

Niu, G.-Y., et al. (2011), The community Noah land surface model with multiparameterization options (Noah-MP): 1. Model description and evaluation with local-scale measurements, J. Geophys. Res., 116, D12109, doi:10.1029/2010JD015139.

We will add a short sentence describing this as follows:

The land component of PWRF is the community Noah land surface model with multi-parameterization options (Noah-MP, Niu et al., 2011) and distinguishes between bare ground, land ice (glaciers) and sea ice. The latest version includes a three-layer snow model (Yang et al., 2003). It improves the representation of surface fluxes, surface meltwater production, percolation and retention/refreezing in the snow layers and surface runoff, and has a refined snow water equivalent reproduction (closer to reality) with an improved diurnal cycle of the snow skin temperature.

The land component of PWRF is the community Noah land surface model with multi-parameterization options (Noah-MP, Niu et al., 2011) and distinguishes between bare ground, land ice (glaciers) and sea ice. The latest version includes a three-layer snow model (Yang et al., 2003) and has been shown to perform well against observations at Col de Porte (French Alps) and Sleeper River Watershed (Vermont, USA) (Niu et al., 2011): It improves the representation of surface fluxes, surface meltwater production, percolation and retention/refreezing in the snow layers and surface runoff, and has a refined snow water equivalent reproduction (closer to reality) with an improved diurnal cycle of the snow skin temperature.

**Line 124: 61 model levels -> 61 vertical levels**

Thank you, we will change 'model levels' by 'vertical levels'.

It has 210 x 240 grid cells and 61 model levels.

It has 210 x 240 grid cells and 61 vertical levels.

**Line 149: "The experiments were set with the aim of testing the model skill...". Similar to my first round of comments, you are not really testing model skill. That is, you are not evaluating the model's ability to reproduce observations or another model. Would suggest rephrasing to "...with the aim of comparing various model configurations for different sea ice cover conditions (winter vs summer)."**

Thank you for this comment, we will adopt your suggestion.

The experiments were set up with the aim of testing the model skill for different sea ice cover conditions (winter versus summer).

The experiments were set up with the aim of comparing various model configurations for different sea ice cover conditions (winter vs summer)

**Line 151: "cast study" -> "case study"**

We will change 'cast study' for 'case study'.

our cast study including the conservation of heat and mass,

our case study including the conservation of heat and mass,

**Line 197: "This standard import way" -> "This standard import method"**

Thank you, we will change 'way' for 'method'.

This standard import way (used in SKRIPS)

This standard import method (used in SKRIPS)

**Line 334-335: "...the components are added individually and the time step non conveective precipitation term is ignored."**

**I think the words "time step" are not needed here? So I think this should be "...the components are added individually and the non-convective precipitation term is ignored."**

This is the name of the variable as is defined in PWRF, but we will remove 'time step' for better readability.

In the P-SKRIPS, the components are added individually and the time step non convective precipitation term is ignored.

In the P-SKRIPS, the components are added individually and the non convective precipitation term is ignored.

**Line 362: P-SRIPS -> P-SKRIPS**

Thank you for spotting this typo, we will replace 'P-SRIPS' by 'P-SKRIPS'

In P-SRIPS, the snow cover

In P-SKRIPS, the snow cover

**Line 398-399: Would suggest changing to "The energy imbalance of up to 922 W m-2 due to non-conservation of energy in the previous Skripps-KAUST version of the model likely affects the heat content ..." or similar.**

Thank you for this comment, we will rephrase the sentence accordingly.

The non-conservation of up to 922 W m$^{-2}$ is likely affecting the heat content of the atmosphere and deep convection of the ocean.

The energy imbalance of up to 922 W m$^{-2}$ due to non-conservation of energy in the previous Skripps-KAUST version of the model likely affects the heat content of the atmosphere and deep convection of the ocean.

**I would also like to note that I am pleased to see that a solution has hopefully been come to for the issue of code availability.**

Thank you, we are also pleased that we could find an agreement with the editor!

Author response to the Referee Comments by Dr Guillaume Boutin on the manuscript:

**Conservation of heat and mass in P-SKRIPS version 1: the coupled atmosphere-ice-ocean model of The Ross Sea**

Alena Malyarenko, Alexandra Gossart, Rui Sun, and Mario Krapp

submitted to Geoscientific Model Developments (https://doi.org/10.5194/egusphere-2022-1135)

We thank the Reviewer Dr Guillaume Boutin for his review of our manuscript and are pleased with his positive comments. Please find the response to each of the comment below. The reviewer's comment is displayed in **bold text**, the reply is shown in normal text, the text from the original manuscript is shown in blue, and proposed change to the manuscript is shown in red.
* * *
**Small typo: L322: 1.1013–> I believe you mean 1.10^13.**

Yes, thank you for spotting this typo.

10    The ocean receives a larger amount of latent heat in the SKRIPS simulation with an almost constant bias of 1.1013 W

The ocean receives a larger amount of latent heat in the SKRIPS simulation with an almost constant bias of $1.10^{13} W$.